# Effects of Drying Process and High Hydrostatic Pressure on Extraction of Antioxidant Ergothioneine from *Pleurotus citrinopileatus* Singer

**DOI:** 10.3390/foods13060878

**Published:** 2024-03-14

**Authors:** Changge Zhang, Yitong Xie, Danyi Liu, Rongxu Liu, Jianchun Han

**Affiliations:** 1College of Food Science, Northeast Agricultural University, Harbin 150030, China; zhangchanggggg@163.com (C.Z.); xyt1439@163.com (Y.X.); 2Heilongjiang Institute of Green Food Science, Harbin 150030, China; danyiliu@163.com

**Keywords:** *Pleurotus citrinopileatus* Singer, high-hydrostatic-pressure extraction, ergothioneine, drying methods, antioxidant

## Abstract

This study evaluated the effects of different drying techniques on the physicochemical properties of *Pleurotus citrinopileatus* Singer (*P. citrinopileatus*), focusing on the ergothioneine (EGT) contents. The *P. citrinopileatus* was subjected to natural ventilation drying (ND), freeze-drying (FD), and hot-air drying (HD). EGT was extracted using high-hydrostatic-pressure extraction (HHPE), and response surface methodology (RSM) was employed with four variables to optimize the extraction parameters. The crude EGT extract was purified by ultrafiltration and anion resin purification, and its antioxidant activity was investigated. The results showed that the ND method effectively disrupted mushroom tissues, promoting amino acid anabolism, thereby increasing the EGT content of mushrooms. Based on RSM, the optimum extracting conditions were pressure of 250 MPa, extraction time of 52 min, distilled water (dH_2_O) as the extraction solvent, and a 1:10 liquid–solid ratio, which yielded the highest EGT content of 4.03 ± 0.01 mg/g d.w. UPLC-Q-TOF-MSE was performed to assess the purity of the samples (purity: 86.34 ± 3.52%), and MS^2^ information of the main peak showed primary ions (*m*/*z* 230.1) and secondary cations (*m*/*z* 186.1050, *m*/*z* 127.0323) consistent with standard products. In addition, compared with ascorbic acid (VC), EGT showed strong free radical scavenging ability, especially for hydroxyl and ATBS radicals, at more than 5 mmol/L. These findings indicate that the extraction and purification methods used were optimal and suggest a possible synthetic path of EGT in *P. citrinopileatus*, which will help better explore the application of EGT.

## 1. Introduction

Ergothioneine (EGT) is a naturally occurring betaine amino acid synthesized in several microorganisms, particularly fungi-like mushroom fruiting bodies and actinomycetes. However, it can only be obtained from plants and animals through the soil and diet [1]. Figure 1A shows that the compound EGT undergoes a tautomeric equilibrium between its thiol and thione forms. However, EGT exists mainly in the thione form at physiological pH and has a high REDOX potential [2,3]. Therefore, EGT is more resistant to auto-oxidation than other thiols (such as glutathione), has good free radical elimination ability [4] and antioxidant activity [5], and is a highly effective antioxidant and cell protectant [6], making it a compound with multiple potential health benefits [7]. Mushrooms are the richest dietary source of EGT in humans [7]. In fungi, the synthesis of ergothioneine depends on the glutathione content and its precursors, and the EGT content varies greatly depending on the strain and growth conditions [8]. Different extraction, synthesis, and purification methods may contribute to the reported EGT concentrations [9,10,11,12].

*Pleurotus citrinopileatus* Singer (Figure 1B) is rich in EGT [13], and its efficient development and utilization hold promise as a potential source of human EGT supplements. Traditional food storage and processing involves drying, and the commonly used methods for edible mushrooms include natural drying, hot-air drying, and freeze-drying [14]. However, the EGT content in mushrooms varies depending on the drying methods, potentially owing to the degrees of influence on the tissue and cell structures of mushrooms [15]. In addition, drying breaks down the proteins into amino acids [16], which are important synthetic substances in EGT [17].

High-hydrostatic-pressure extraction (HHPE) involves applying 100–1000 MPa hydrostatic pressure to the material and solvent under normal temperature conditions, holding the pressure for a period, and reducing the pressure to normal pressure in a short time after holding the pressure. A large pressure difference between inside and outside the cell will quickly transfer the active ingredients in the material to the extraction agent, thus completing the high-pressure extraction process [18]. Recently, high hydrostatic pressure (HHP) has found widespread application in the extraction of naturally active substances from mushrooms and other plants. For example, natural melanin has been extracted from the wild mushroom *Auricularia auricula* [19], bioactive peptides have been extracted from mushrooms [20], and phenolic compounds have been extracted from various plants [18]. Moreira [21] showed that HHPE is more effective than atmospheric pressure extraction, with an increase in antioxidant activity measured by fluorescence recovery after photobleaching FRAP, DPPH, and ABTS assays (29%, 48%, and 70%) and an increase of approximately 40% for all compounds. HHP increases the extraction rate of bioactive compounds and can be performed in a relatively short period. Despite the extensive use of HHP in the extraction of various substances, no report exists on the extraction of EGT from *P. citrinopileatus* using HHP.

Moreover, the effects of different drying methods on the mushrooms’ tissue structure, cell structure, and physicochemical properties, and the variation in the mushroom EGT content, have been less studied. Therefore, we analyzed the effects of different drying methods on the properties of *P. citrinopileatus* and its EGT content. The extraction process of EGT under ultrahigh pressure was optimized using the response surface method. The crude extract was purified using an ion-exchange resin to obtain high-purity EGT. Finally, the antioxidant activity of different concentrations of EGT was studied.

## 2. Materials and Methods

### 2.1. Materials

Fresh *P. citrinopileatus* was purchased from Biobw Biotechnology Co., Ltd. (Beijing, China). Ethanol, formic acid, and acetonitrile were obtained from Sinopharm Chemical Reagent Co., Ltd. (Shanghai, China). L-(+)-ergothioneine, mixed amino acid standard, and ascorbic acid (VC) were obtained from Shanghai Yuanye Bio-Technology Co., Ltd. (Shanghai, China). DPPH-free radical scavenging capacity assay kits (spectrophotometer), hydroxyl-free radical scavenging capacity assay kits (spectrophotometer), and ABTS-free radical scavenging capacity assay kits were purchased from Solarbio Life Sciences (Beijing, China). HCN-5 and TSF-1 resins were purchased from Tianjin Yunkai Resin Technology Co., Ltd. (Tianjin, China). The other substances were all analytically graded.

### 2.2. Analysis of Different Drying Methods on EGT of P. citrinopileatus 

#### 2.2.1. Drying Process of *P. citrinopileatus*

Natural ventilation drying (ND): Fresh *P. citrinopileatus* was obtained, kept in a ventilated place at 20–22 °C, and allowed to dry naturally for 72 h until it reached a constant weight. Subsequently, the ND mushrooms were broken up with a grinder (IKA MuItiDrive BASIC S025, IKA (Guangzhou) Instrument Equipment Co., Ltd., Guangzhou, China) and sifted through a 50-mesh screen. The mushroom powder was sealed and stored in a cool and dry place for further use.

Freeze drying (FD): Fresh *P. citrinopileatus* was subjected to pre-freezing at −20 °C for 3 h and then placed in the drying room of the freeze-drying machine (Pilot7-12E, BIOCOOL Instrument Co., Ltd., Beijing, China) for freeze-drying for 24 h until the samples reached a constant weight. The FD mushrooms were then broken up using a grinder (IKA MuItiDrive BASIC S025, IKA (Guangzhou) Instrument Equipment Co., Ltd., China) and sifted through a 50-mesh screen. The mushroom powder was sealed and stored in a cool and dry place for further use.

Hot-air drying (HD): Fresh *P. citrinopileatus* was placed in a natural convection drying oven (DHG-9015A, Yiheng Scientific Instrument Co., Ltd., Shanghai, China) at 60 °C and dried for 24 h until a constant weight was achieved. The HD mushrooms were then broken up using a grinder (IKA MuItiDrive BASIC S025, IKA (Guangzhou) Instrument Equipment Co., Ltd., China) and sifted through a 50-mesh screen. The mushroom powder was sealed and stored in a cool and dry place for further use.

#### 2.2.2. Amino Acid Analysis

The change in the amino acid content of the mushrooms in different states (fresh, ND, FD, and HD) was determined and its influence on the EGT content was analyzed [22]. A specific amount of mushroom in the different states (fresh, ND, FD, and HD) was weighed into hydrolysis tubes, mixed with a 10 mL 1:1 hydrochloric acid solution, and placed in a 110 °C oven for hydrolysis for 22 h. Following hydrolysis, the samples were cooled, transferred to a 10 mL volume bottle, and made up to a final volume of 10 mL with 0.02 mol/L HCL. The sample solution was filtered through a 0.22 μm membrane. Then, an aliquot of 20 μL of the mixed amino acid standard working solution and sample determination solution was injected into the amino acid analyzer (LA8080, HITACHI (Shanghai) International Trade Co., Ltd., Shanghai, China). Seventeen amino acids were used to generate the standard curves.

#### 2.2.3. Scanning Electron Microscope (SEM) 

The mushrooms in different states were cut into 2 mm thick slices with a blade, fixed with 2% glutaraldehyde for 10 h, and dehydrated with a 50%, 70%, 90%, and 100% ethanol solution thrice; the medium was replaced with tert-butanol, and the samples were freeze-dried for 5 h. Finally, the tissue structure of the samples was observed using scanning electron microscopy and images were obtained [23] at 500×, 1000×, and 1500× magnifications.

#### 2.2.4. Transmission Electron Microscope (TEM)

A TEM was used to observe the tissue (5 mm × 5 mm × 1 mm) cut from the cap of the mushrooms in different states (fresh, ND, FD, and HD) [24]. First, the tissues were kept in a 2 mL centrifuge tube, and 0.1 M phosphate buffer (pH 7.2, containing 5% glutaraldehyde solution) was added. The samples were vacuumed for 10 min, vented until the plant tissues sank to the bottom of the centrifuge tube, and then fixed at 4 °C for 24 h. The fixed tissue was rinsed several times with 0.1 M phosphate buffer (pH 7.2) and then fixed in 1% osmium tetroxide for 1–2 h. The samples were then removed, rinsed with phosphate buffer several times, eluted with gradient concentrations of ethanol (30%, 50%, 70%, 80%, and 95%), embedded in acetic acid resin, sliced, and, finally, observed and photographed by TEM at 5000× and 10,000× magnification.

#### 2.2.5. EGT Content Determination

The contents of EGT in the mushrooms in different states (fresh, ND, FD, and HD) were determined. The moisture contents of mushrooms in different states (fresh, ND, FD, and HD) were determined by a rapid moisture meter (HX204, METTLER TOLEDO Technology Co., Ltd., Guangzhou, China) and were 90.32%, 0.77%, 0.52, and 0.56%, respectively. Subsequently, the samples were treated according to the existing method [25]. The fresh mushrooms or mushroom powders (ND, FD, and HD) were mixed with 60% ethanol at a solid-to-liquid ratio of 1:30 (*w*/*v*). The solid–liquid mixture was extracted in a water bath maintained at a constant temperature of 70 °C for 1 h and centrifuged at 9500× *g* for 10 min. The supernatant was collected, and the volume was made up to 100 mL with 60% ethanol to determine the amount of EGT extracted by HPLC (1260 II Prime, Agilent Technologies (China) Inc., Beijing, China) [25]. 

HPLC detection conditions: The column (XBridge^®^ BHE Amide 5 μm) was manufactured by Agilent Technologies Inc. The mobile phase was 98:2 water and methanol, flow rate was 1 mL/min, sample volume was 20 μL, and detection wavelength was 254 nm. EGT yields were calculated using Equation (1).
(1)$$EGT\;contents(\mathrm{mg}/g\;d.w.)=\frac{C\times V}{m\times (1-w)\times 1000}$$
where *C* is the EGT content of EGT by HPLC, *V* is the constant volume, *m* is the weight of the mushrooms, and *w* is the moisture content of the mushrooms.

### 2.3. EGT Extraction Using High Hydrostatic Pressure (HHPE)

According to the above experiment, the EGT content in ND mushrooms was the highest using traditional extraction methods. Therefore, we chose ND mushrooms as raw materials and used HHP as the extraction method to further optimize the extraction process.

The bag containing ND mushroom powder and the extraction solvent as a sample was placed in the pressure tank. The samples were subjected to an HHP instrument (HHP-400 MPa, RenHe Electromechanical Engineering Co., Shenyang, China). For single-factor experiments, the extraction pressure, time, and liquid–solid ratio for single-factor studies varied from 0.1 to 400 MPa, 20 to 60 min, and 1:1 to 1:20 (*v*/*w*), respectively. The extraction solvents used were dH_2_O and 20%, 40%, 60%, and 80% ethanol, respectively. The sample was centrifuged at 9500× *g* following extraction. The crude EGT extract was prepared using the supernatant.

### 2.4. Optimization of EGT Extraction

The ND mushrooms were used as raw materials. The link between the extraction conditions and the EGT yield (Y) was studied using the Box–Behnken design (BBD) based on the single-factor experiment in order to determine the best combination of different factors. The fitting study was conducted using Design-Expert 13 software. Four factors were studied as independent variables: extraction pressure (A), extraction time (B), extraction solvent (C), and liquid–solid ratio (D) (Table 1).

### 2.5. Purification of EGT

Using the optimized conditions of HHPE obtained in Section 2.4, we extracted EGT from the ND mushrooms and obtained the crude extract. The crude extract was subjected to microfiltration (400 nm micro-filtration membrane) and ultrafiltration (4 kDa ultrafiltration membrane) step by step to remove macromolecular substances. The filtrate was collected and concentrated (240× *g*, 22 °C) for 4 h using a vacuum centrifugal concentrator (Concentrator plus, Eppendorf (China) Inc., Shanghai, China). The concentrated liquid was collected for ion exchange chromatography. The bubbles were exhausted after injecting the prepared HCN-5 and TSF-1 resins into a glass chromatographic column. Distilled water was added for several hours to equilibrate and compact the resin in the column. The concentrated liquid was diluted to 20 mg/mL with dH_2_O and injected into the column using a digital display pump at a constant flow rate of 2 BV/h. After the resin adsorption saturation, the sample was eluted using 20 times the volume of dH_2_O to obtain eluent. The eluent was collected and freeze-dried to obtain a freeze-dried powder for quantitative and qualitative analysis using UPLC-Q-TOF-MS^E^ (Q Exactive Plus, Thermo Fisher Scientific (China) Inc., Shanghai, China) [26].

Sample treatment: A sample weighing 0.01 g (accurate to 0.00001 g) was added to a 10 mL volume bottle, to which 70% acetonitrile was added and mixed well until the sample was completely dissolved. Subsequently, the volume was adjusted to the scale, diluted to the corresponding ratio, passed through a 0.22 μm filter membrane, and shaken on a shaker. 

Detection conditions: Aqueous 0.1% formic acid (A) and acetonitrile (B) were used as the mobile phase. Gradient elution was performed as follows: 0–1.5 min, 5% B; 1.6–3.5 min, 10% B; 3.6–5.0 min, 5% B. The loading quantity and speed were 2 μL and 0.3 mL/min, respectively. Using an ESI source, EGT was identified by MS/MS. A positive ionization mode electrospray source with a range of 50–255 *m*/*z* was used for MS. EGT was identified based on the ionic group, retention time, and peak area. The EGT standard was used as an external standard, and the EGT was quantified according to the standard curve y = 320.287 x + 359.74 (R^2^ = 0.9997) in Figure 2, where y represents the response and x represents the EGT concentration (ng/mL).

### 2.6. Measurement of Antioxidant Activities of EGT

#### 2.6.1. DPPH Free Radical Scavenging Activity

The DPPH•-scavenging ability of EGT at different concentrations was assessed according to previous reports [3,9]. Purified EGT was diluted with distilled water to 10, 5, 2, 0.5, and 0.1 mmol/L concentrations. Free radical clearance was determined using an assay kit (Yuanye Bio-Technology Co., Ltd., Shanghai, China) [27]. Absorbance was measured at 515 nm using a UV spectrophotometer. Ethanol served as the blank and ascorbic acid served as the standard. The DPPH free radical scavenging (%) was evaluated using Equation (2).
(2)$$The\;DPPH\;free\;radical\;scavenging\;(\%)=(1-\frac{A_{1}-A_{2}}{A_{0}})\times 100\%$$
where *A*_0_ is the absorbance of the DPPH solution alone, *A*_1_ is the absorbance of the samples mixed with DPPH, and *A*_2_ is the samples’ absorbance in ethanol devoid of DPPH.

#### 2.6.2. ABTS Free Radical Scavenging Activity

The ABTS free radical scavenging capacity of EGT at different concentrations (10, 5, 2, 0.5, and 0.1 mmol/L) was measured at 405 nm using an assay kit (Yuanye Bio-Technology Co., Ltd., Shanghai, China) [28]. The positive control used VC. ABTS free radical scavenging activity (%) was evaluated using Equation (3).
(3)$$The\;ATBS\;free\;radical\;scavenging\;(\%)=(1-\frac{A_{1}-A_{2}}{A_{0}})\times 100\%$$
where *A*_0_ is the absorbance of the ATBS solution alone, *A*_1_ is the absorbance of the samples mixed with ATBS, and *A*_2_ is the samples’ absorbance in ethanol without ATBS.

#### 2.6.3. Hydroxyl Radical Scavenging Activity

Purified EGT was diluted with distilled water to 10, 5, 2, 0.5, and 0.1 mmol/L concentrations. The hydroxyl radical scavenging ability of EGT at various concentrations was measured at 536 nm using an assay kit (Yuanye Bio-Technology Co., Ltd., Shanghai, China) [29]. VC was used as the positive control. Hydroxyl radical scavenging activity (%) was evaluated using Equation (4).
(4)$$The\;hydroxyl\;radical\;scavenging\;(\%)=(\frac{A_{1}-A_{2}}{A_{0}-A_{2}})\times 100\%$$
where *A*_0_ is the absorbance of the blank, *A*_1_ is the absorbance of the sample, and *A*_2_ is the absorbance of the contrast agent.

### 2.7. Statistical Analysis

IBM SPSS Statistics 23 (IBM Corp., Armonk, NY, USA) and Origin 21.0 (OriginLab Corp., Northampton, MA, USA) were used to analyze the experimental data. The mean ± standard deviation was used to represent the outcomes of three independent replicate tests. Statistical significance was set at *p* < 0.05.

## 3. Results

### 3.1. Effects of Drying Techniques on the Extraction of EGT from P. citrinopileatus

Ergothioneine (EGT) content is influenced by enzyme, oxygen, and prerequisite amino acids [30]. Therefore, we studied the effects of different drying methods on the amino acid content and tissue structure of *P. citrinopileatus* and EGT yield to explore whether the changes in tissue structure and prerequisite amino acids affected the EGT content. 

Figure 3 shows that the yield of EGT was the highest in the ND-treated mushrooms (3.72 ± 0.06 ^A^ mg/g d.w., *p* < 0.05). The EGT yields of fresh mushrooms (2.63 ± 0.06 ^B^ mg/g d.w.) and FD-treated mushrooms (2.53 ± 0.04 ^B^ mg/g d.w.) were lower than those of ND-treated mushrooms. The yield of EGT in HD-treated mushrooms was the lowest (2.37 ± 0.07 ^C^ mg/g d.w.). These findings suggest that, compared to the fresh state, ND treatment can significantly increase the content of EGT in mushrooms, FD treatment shows no significant difference, and HD treatment can significantly reduce the EGT content of mushrooms.

Numerous studies have shown that glutathione, cysteine, histidine, and methionine are crucial precursors affecting EGT synthesis [1]. Therefore, we determined the amino acid contents of mushrooms in different states. As shown in Table 2, the histidine contents of the mushrooms in different states (fresh, HD, FD, and ND) were 2.60 ± 0.01 ^a^, 2.28 ± 0.005 ^b^, 2.23 ± 0.005 ^b^, and 2.16 ± 0.01 ^b^, respectively. The contents showed a downward trend and were negatively correlated with EGT content. The fungal synthesis pathway is shown in Figure 4. Studies have shown that the fungal L-EGT synthesis pathway involves the transfer of three methyl groups from S-adenosylmethionine (SAM) to histidine to form histidine trimethylenolide by the synthetase Egt1. Oxygen and cysteine are then used to catalyze histidine trimethylenolide to form heptylenocysteine sulfoxide and, finally, L-EGT is synthesized by the Egt2 synthetase enzyme [30,31]. Histidine and oxygen are the key components of this pathway [32,33]. Therefore, we speculate that one of the reasons for the high EGT content under ND conditions is that natural drying takes the most time, and the water activity is high most of the time (for approximately the first 10 h, the water activity is higher than 50%). Thus, the key synthetases can remain active for a long time and can make sufficient use of oxygen to catalyze the precursor amino acid (histidine) to synthesize EGT, resulting in a reduction in the histidine content. In addition, in the freeze-drying process, the activity of the enzyme is inhibited, and there is a lack of oxygen. Thus, the degree of synthesis is no different from that in the fresh state, so the EGT content is not significantly different.

SEM can reflect the microstructure of mushrooms in different states (Figure 5). Fresh mushrooms exhibit dense and compact tissues, with no porosity between the tissues, and possess excellent structural integrity. As water evaporates, capillary tubes produce contraction stress, causing mushroom tissues to contract differently [25]. The water loss is relatively gentle with ND treatment, preserving the cell structure and resulting in a well-maintained organizational structure and uniform distribution [34]. Low temperatures induce the formation of large ice crystals in mushroom tissues, puncturing the cell, and destroying the cell structure [35,36]. High temperatures cause rapid water loss in mushrooms, destroying more cellular structures and resulting in the collapse of tissue structures, affecting nutrient conservation.

As shown in Figure 6, the internal microstructure of the cell was observed using TEM. Typically, the cell wall constitutes approximately 30% of the dry matter content of fungal cells, providing structural support, maintaining cell shape and hardness, and controlling the adhesion between cells. Changes in cell wall structure and intracellular components directly affect amino acid synthesis [36]. FD and HD treatments caused a rapid loss of water in mushroom cells from the inside to the outside, increasing the thickness of the pectin layer between the cell walls, resulting in pronounced lignification of fibers, reduced elasticity, apparent cell shrinkage, and deformation, which is not conducive to the synthesis of EGT inside and outside the cell [37]. In contrast, the ND mushrooms exhibited a uniform distribution of cells, initiating plasmolysis, which is conducive to EGT synthesis, thereby increasing the EGT yield.

### 3.2. Single-Factor Experiments for EGT

The ND mushrooms were used as raw materials. The yield of EGT extraction was examined in relation to the effects of extraction pressure, extraction time, solvent, and liquid–solid ratio (Figure 7). When the extraction pressure was increased from 0.1 to 400 MPa, the EGT yield increased significantly (Figure 7A, *p* < 0.05), which can be attributed to the drastic pressure difference generated by the instantaneous pressure boost, allowing the solvent to break the cell and tissue structures of mushrooms to extract the active components more effectively [38]. The greater the pressure difference, the greater the impact on the extraction effect [39]. According to Fick’s second law [40], pressure difference can be used as the force of the extracted substance, and the greater the pressure difference, the stronger the force and the more extract can be obtained. At the same time, the quality of the extracted material is also related to the extraction time. Moreover, studies have shown that pressures below 600 MPa do not destroy the structures of the extracted substances [41]. The EGT yield was the highest (4.01 ± 0.08 ^a^ mg/g d.w.) at 300 MPa. Owing to the same degree of cell membrane disruption caused by 300 and 400 MPa, there was no discernible change in the EGT yield (*p* > 0.05).

Figure 7B shows that the yield of EGT increased as the extraction time increased (*p* < 0.05). When the extraction time increased from 20 min to 50 min, the EGT yield increased from 2.35 ± 0.05 ^c^ mg/g d.w. to 4.09 ± 0.07 ^a^ mg/g d.w. The extended duration of pressure allowed more cells to break and ensured enhanced contact between the material and the extraction solvent, facilitating the complete dissolution of EGT in water and thereby increasing the EGT yield [42]. However, when the time was further extended from 50 min to 60 min, the increasing trend became more gradual, and the difference was not statistically significant as the mass transfer of the extractant and the active ingredient reached a balance within a specific time [43].

The effects of the extraction solvent on EGT yield are shown in Figure 7C. Traditional extraction methods require organic solvents to reduce the protein and polysaccharide contents in the extraction solution, thereby increasing the yield of EGT [44]. However, there was no significant difference when distilled water (3.94 ± 0.07 ^a^ mg/g d.w.) and 60% ethanol (4.03 ± 0.03 ^a^ mg/g d.w.) were used as solvents for HHP treatment (*p* > 0.05). The lack of difference could be attributed to the inherent water solubility of EGT itself [2] and the ultrahigh pressure, which efficiently transferred the water solvent into the cell, increasing EGT dissolution. Simultaneously, the substantial pressure difference caused the cell wall to break, causing the overflow of the EGT-containing aqueous solution [45]. 

As shown in Figure 7D, the EGT yield increased as the liquid–solid ratio increased from 1:1 to 1:12 and peaked at 1:10 (*p* < 0.05), with a yield of 4.04 ± 0.08 mg/g d.w. Higher leaching rates may result from an increased diffusion rate with increasing solvent concentration, favoring the dissolution of bioactive components [38]. However, following the peak at 1:10 *v*/*w*, the EGT yield showed no discernible decrease and remained stable; therefore, the ratio was set to 1:10 *v*/*w*.

### 3.3. Optimization Experiments for EGT

The Box–Behnken design is helpful for response structure methodology (RSM) as it avoids conducting too many exploratory experiments, is based on single-factor trials, and offers a scale of experimental variables [29]. The single-factor experiment indicated that 27 runs of the three variables were conducted to identify the ideal extraction conditions for EGT. The four variables were the extraction pressure (A), extraction time (B), extraction solvent (C), and liquid-to-solid ratio (D). The experimental parameters and EGT yields (1.23–4.17 mg/g d.w.) are listed in Table 3. By performing a multiple regression analysis of the experimental values, Equation (1) of the response model for the EGT yield (Y) was obtained as follows:Y (mg/g) = 4.12 − 0.5100A − 0.2125B − 0.0292C + 0.0100D − 0.5925AB − 0.1700AC − 0.1425AD − 0.3325BC − 0.0125BD − 0.1050CD − 0.5683A^2^ − 0.9621B^2^ − 0.9821C^2^ − 0.4808D^2^


Table 4 and Figure 8 illustrate that the impacts of A, B, C, D, AB, AC, AD, BC, BD, CD, A^2^, B^2^, C^2^, and D^2^ on the EGT extraction were significant (*p* < 0.05). The F-value was used to assess the effects of various parameters on EGT extraction. The effectiveness of EGT extraction was better when the F-value was larger. R^2^ indicates the degree of model fit and the closer R^2^ is to 1, the better the fit. R^2^Adj indicates that R^2^ is modified to avoid overfitting. R^2^pred means the degree to which the model predicts the response of new observations. Models with larger predictive R^2^ values also have better predictive power. The gap between R^2^Adj and R^2^pred should be less than 0.2 [29]. The extent to which each component influenced the EGT extraction was A (extraction pressure) ˃ B (extraction time) ˃ C (extraction solvent) ˃ D (liquid-solid ratio). After the regression equation was improved, the following parameters were found to be ideal for extracting EGT: extraction pressure of 217.5 MPa, extraction period of 51.6 min, liquid-to-solid ratio of 1:9.35, and distilled water (dH_2_O) as the extraction solvent. The predicted EGT yield was 4.197 mg/g d.w. The changed extraction conditions were as follows for operational convenience: extraction pressure of 250 MPa, extraction time of 52 min, dH_2_O as the extraction solvent, and a 1:10 liquid–solid ratio. Under the modified circumstances, the model’s validity was verified and contrasted with that of the conventional solvent extraction (TSE) approach. In three consecutive trials, the measured EGT yield obtained by HHPE (4.03 ± 0.01 mg/g d.w. *p* > 0.05) showed no significant differences from the predicted value, whereas the yield obtained by TSE was 3.37 ± 0.02 mg/g d.w., approximating the theoretical value predicted by RSM. These results demonstrated that HHPE was significantly superior to TSE and that the ideal HHPE conditions for EGT extraction from *P. citrinopileatus* were achievable.

### 3.4. Preparation of High-Purity EGT

According to the optimized extraction conditions, the HHPE crude extract was obtained from ND mushroom powder. The crude extract was separated and purified. As shown in Table 5, after microfiltration and ultrafiltration, the protein and polysaccharide sedimentation rates were 80.86% and 74.34%, respectively. These observations indicate that ultrafiltration could effectively remove macromolecular components, such as proteins and polysaccharides, from the extract and had little effect on EGT preparation [46].

The crude EGT extract was filtered, separated, and purified in series using a canal-cation resin. Figure 9 shows the acquisition time and mass-to-charge ratio of the standard. The purified sample was subjected to UPLC-Q-TOF-MSE quantitative and qualitative analyses, and the results are shown in Figure 10. The concentration was purified by HCN-5 resin and TSF-1 resin in series. The calculated purity was 86.34 ± 3.52% and the main peak retention time was 0.936 min (Figure 10B). In the positive ion scanning mode, the MS_2_ information of the main peak showed (Figure 10C) that primary ions (*m*/*z* 230.1) and secondary cations (*m*/*z* 186.1050, 127.0323) were detected. The spectra of the EGT standard products were consistent (Figure 9). Ion-exchange column chromatography is a method for separating EGT from impurities using ion exchange [10]. EGT is an amphoteric compound with imidazole structural characteristics. Drawing on the structural properties of histidine, researchers believe that EGT has a certain adsorption capacity for cations and anions; therefore, using cationic resin series can effectively improve the purity of EGT [1,2].

### 3.5. Antioxidant Properties of EGT

The antioxidant activity of the EGT purified product obtained in Section 3.4 was determined. In this study, three indicators (DPPH·, ABTS+, and OH· assays) were used to evaluate the antioxidant activity of samples with different concentrations, and ascorbic acid (VC) was used as a control. As shown in Figure 11, the DPPH, ABTS, and hydroxyl radical scavenging activities were positively correlated with concentration. With increased sample concentration, the free radical scavenging rates of VC and EGT increased and stabilized. The free radical scavenging capacity of EGT increased rapidly in the concentration gradient range of 0.1 to 2 mmol/L (*p* < 0.05). In this concentration gradient range of 0.1–10 mmol/L, the DPPH scavenging capacity of VC was higher than that of EGT (Figure 11A). At a concentration of 0.5 mmol/L, the scavenging rate of DPPH· free radicals by VC reached 97.39%. Moreover, when the VC concentration exceeded 0.5 mmol/L, the increase in VC concentration had no significant effect on the DPPH free radical scavenging activity (*p* > 0.05). However, the concentration-dependent nature of EGT was attributed to its DPPH-scavenging capacity. Similar findings were observed by Bao [47], indicating a positive correlation between EGT concentration and its ability to scavenge DPPH free radicals effectively. In the concentration gradient range of 1–10 mmol/L, the scavenging abilities of EGT and VC were not significantly different (*p* > 0.05). When the concentration exceeded 5 mmol/L, the scavenging rate was stable above 90% (Figure 11B). However, the hydroxyl radical scavenging rate of EGT was significantly higher than that of VC in the concentration range of 0.1 to 10 mmol/L (*p* < 0.05). When the concentration was >2 mmol/L, the scavenging rate was stable above 92% (Figure 11C). The results show that EGT can scavenge free radicals through its high REDOX potential [48]; potent free radical scavenging ability, especially its hydroxyl radical scavenging ability; and strong antioxidant activity. Xiong’s and Franzoni’s [49,50] results show that EGT more strongly removes peroxy free radicals (ROO·), hydroxyl free radicals (OH·), and peroxynitrites (ONOO·) than traditional antioxidants.

## 4. Conclusions

We demonstrated that different drying methods decreased the amino acid content in mushrooms and at the same time caused different degrees of contraction and damage to the tissue structure of mushrooms. In the experimental results, under different treatment conditions, the decreasing trend of histidine as a precursor substance was negatively correlated with the increasing trend of EGT content, and the content of EGT was the highest under ND treatment with sufficient oxygen. Combined with the synthetic pathway of EGT in fungi (Figure 4), we speculated that ND conditions were conducive to the synthesis of EGT in fungi. EGT extraction by a high-hydrostatic-pressure method was studied. The effects of pressure, time, solvent, and solid–liquid ratio on extraction were investigated to realize the efficient extraction of EGT from *P. citrinopileatus*, and response surface methodology was used to optimize the extraction. The results show that pressure and time are the most effective parameters affecting the response. The optimal extraction conditions were extraction pressure of 250 MPa, extraction time of 52 min, dH_2_O as the extraction solvent, and a liquid–solid ratio of 1:10. The extraction rate of EGT could be significantly improved by ultra-high pressure. The crude extract of EGT was purified by ultrafiltration and anion resin in series, and high-purity EGT (86.34%) was obtained by UPLC-Q-TOF-MS^E^ identification. EGT from *P. citrinopileatus* possessed a strong reducing power, as well as scavenging capacity for DPPH, ATBS, and hydroxyl free radicals. In conclusion, this study provides a new method for the high extraction rate of EGT with antioxidant properties so that it can be used in food applications.

## Figures and Tables

**Figure 1 foods-13-00878-f001:**
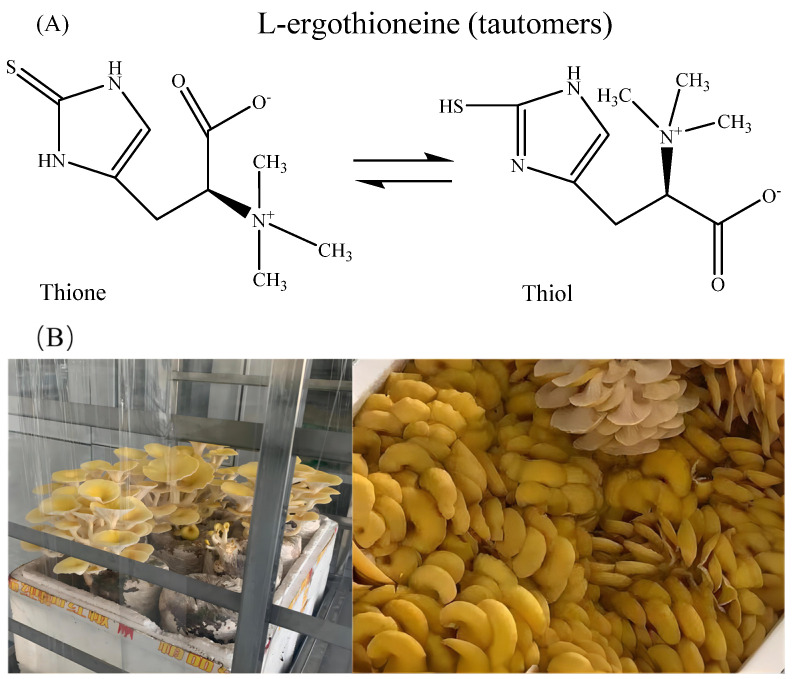
(**A**) Chemical structure of ergothioneine (EGT) thione-thiol tautomers. (**B**) Photograph of *Pleurotus citrinopileatus* Singer mushrooms.

**Figure 2 foods-13-00878-f002:**
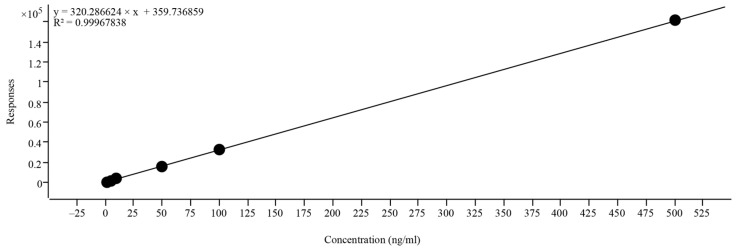
EGT concentration standard curve.

**Figure 3 foods-13-00878-f003:**
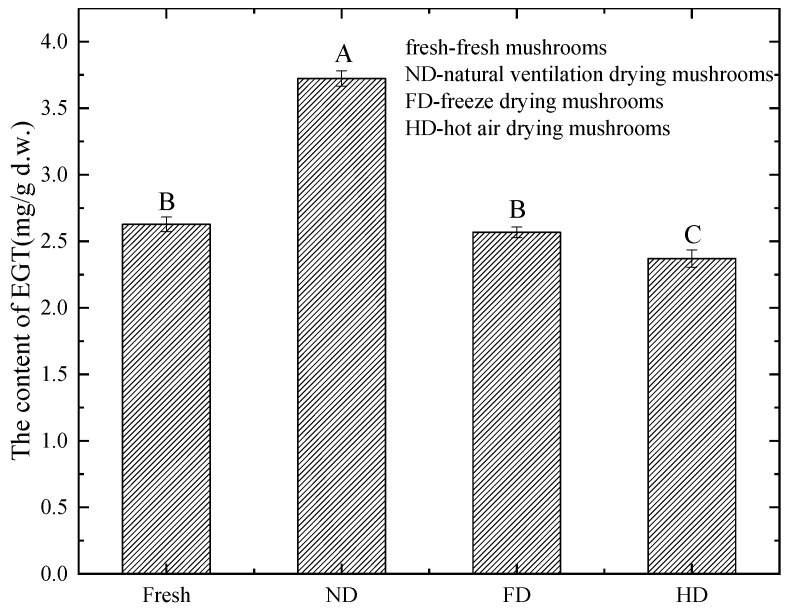
Effects of drying techniques on the extraction of EGT from *P. citrinopileatus*. The data are shown as the mean values (*n* = 3) ± SD. A significant difference (*p* < 0.05) is represented by different letters in the bars.

**Figure 4 foods-13-00878-f004:**
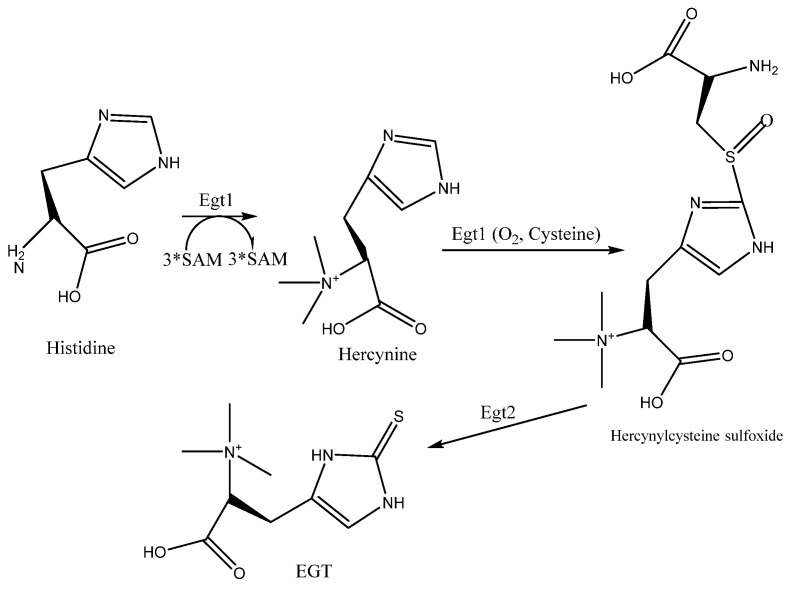
EGT synthesis pathway in fungi.

**Figure 5 foods-13-00878-f005:**
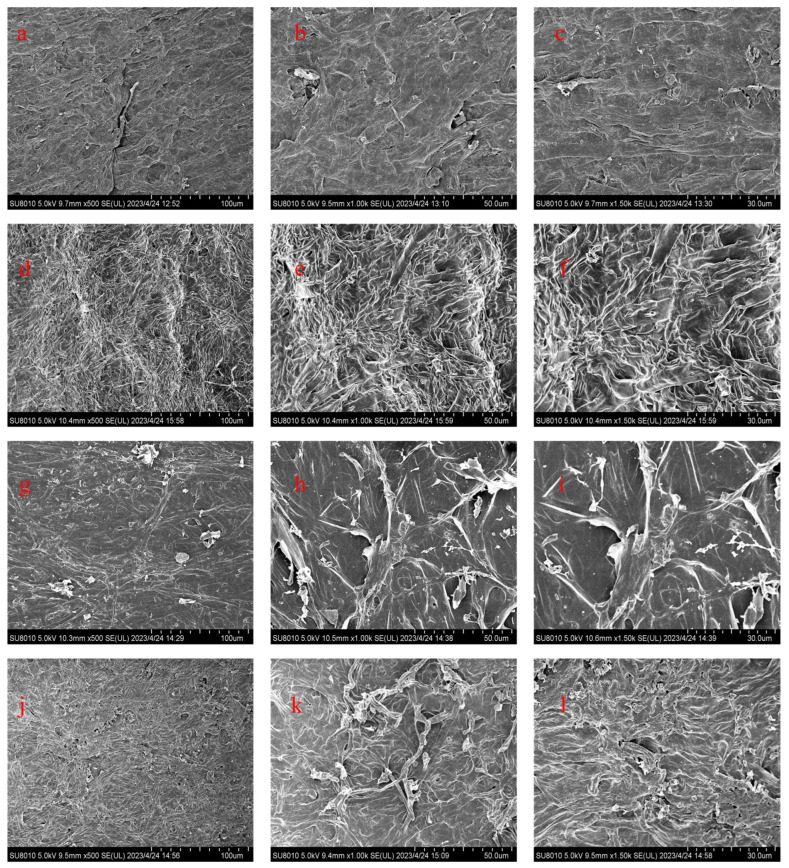
SEM image of *Pleurotus citrinopileatus* Singer. in different states: (**a**–**c**) fresh mushrooms; (**d**–**f**) natural-ventilation-dried (ND) mushrooms; (**g**–**i**) freeze-dried (FD) mushrooms; (**j**–**l**) hot-air-dried (HD) mushrooms. In alphabetical order, magnification is 500×, 1000×, and 1500×.

**Figure 6 foods-13-00878-f006:**
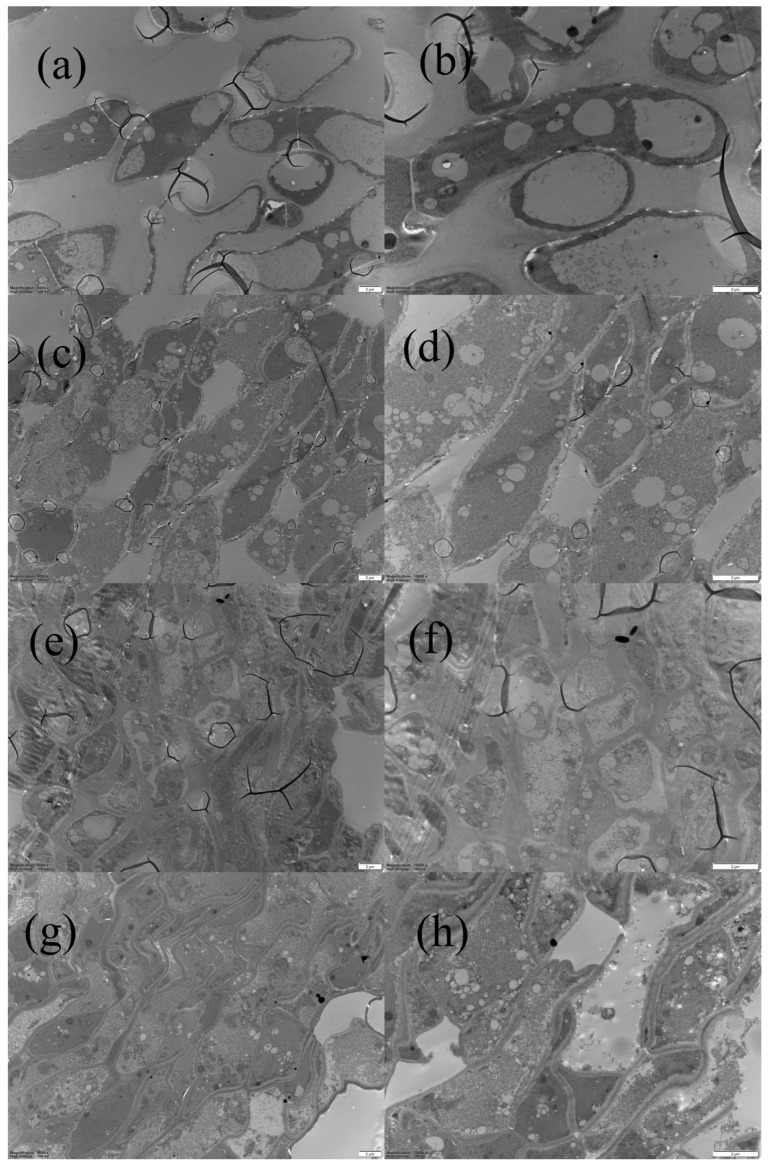
TEM of *Pleurotus citrinopileatus* Singer in different states: (**a**,**b**) fresh mushrooms; (**c**,**d**) natural-ventilation-dried (ND) mushrooms; (**e**,**f**) freeze-dried (FD) mushrooms; (**g**,**h**) hot-air-dried (HD) mushrooms. In alphabetical order, magnification is 5000× and 10,000×.

**Figure 7 foods-13-00878-f007:**
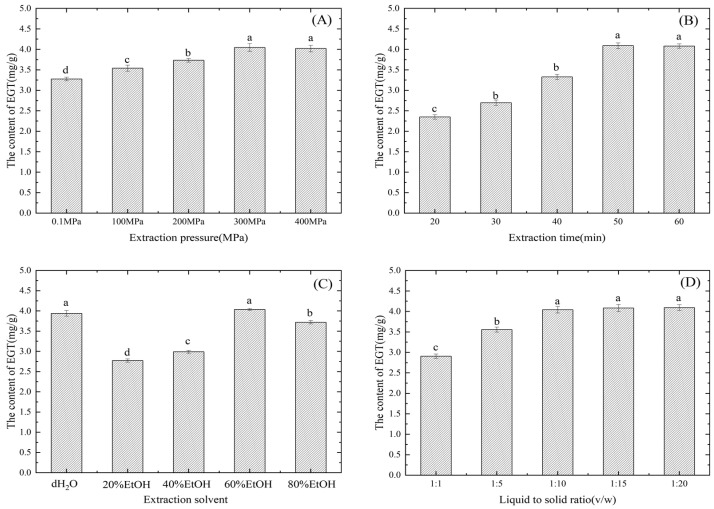
Impact of several conditions on the EGT yield of *Pleurotus citrinopileatus* Singer. The effect of (**A**) extraction pressure, (**B**) extraction time, (**C**) extraction solvent, and (**D**) liquid-to-solid ratio. Data are shown as the mean values (*n* = 3) ± standard deviation. To indicate a significant difference (*p* < 0.05), different letters are used in the bars.

**Figure 8 foods-13-00878-f008:**
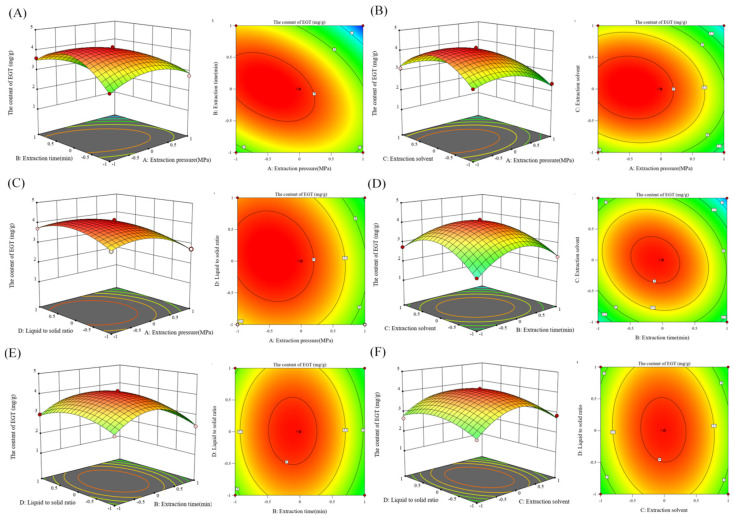
3D and contour plots from BBD. (**A**) Extraction pressure A and extraction time B. (**B**) Extraction pressure A and extraction solvent C. (**C**) Extraction pressure A and liquid–solid ratio D. (**D**) Extraction time B and extraction solvent C. (**E**) Extraction time B and liquid–solid ratio D. (**F**) Extraction solvent C and liquid–solid ratio D.

**Figure 9 foods-13-00878-f009:**
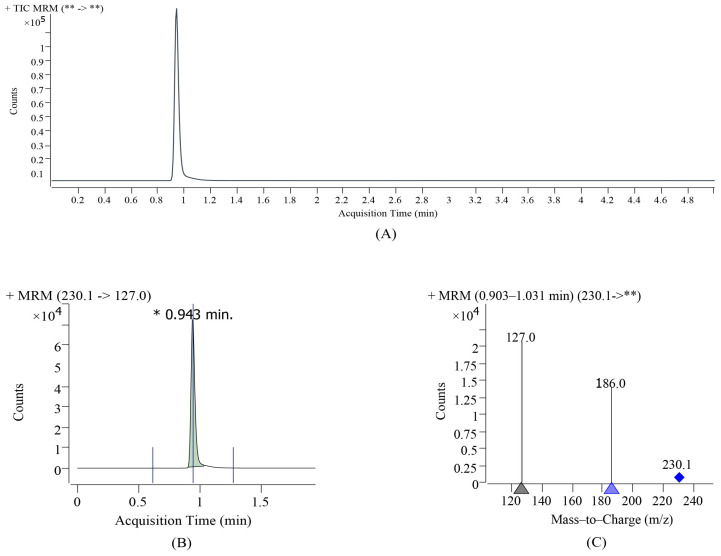
Atlas of the EGT standard. (**A**,**B**) Liquid chromatogram for product determination. (**C**) The mass spectrum of the main peak.

**Figure 10 foods-13-00878-f010:**
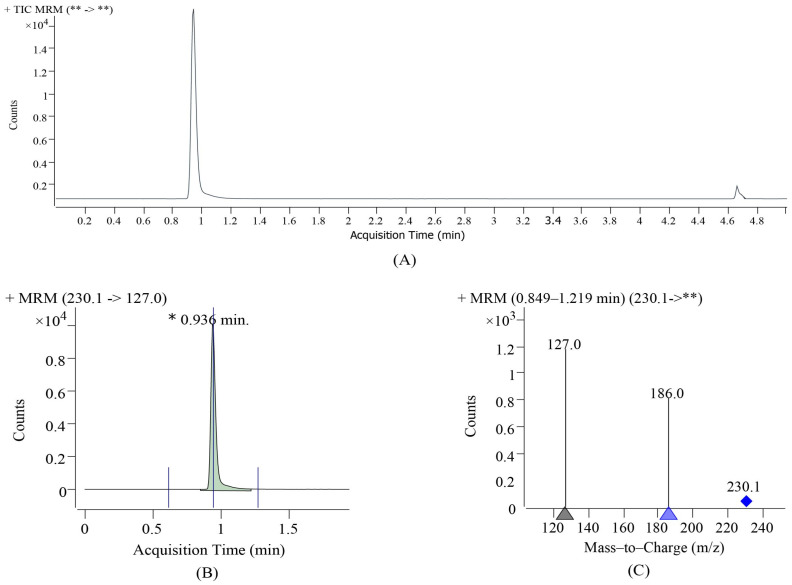
Atlas of EGT purified sample. (**A**,**B**) Liquid chromatogram for product determination. (**C**) The mass spectrum of the main peak.

**Figure 11 foods-13-00878-f011:**
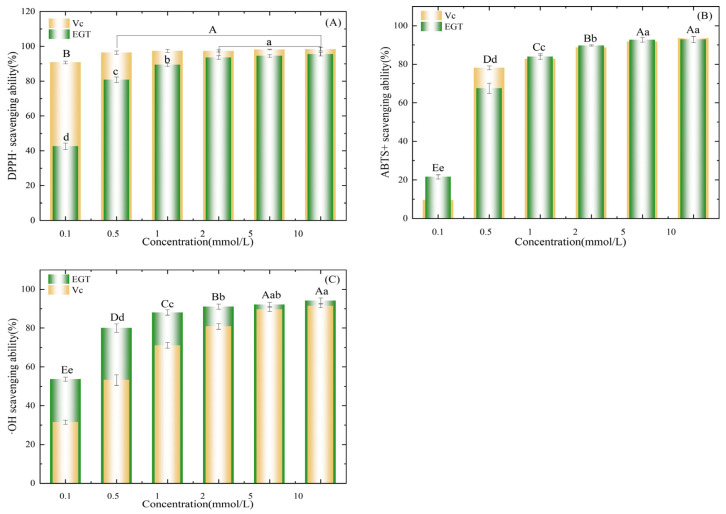
Antioxidant capacity of EGT. (**A**) DPPH free radical scavenging rate. (**B**) ABTS free radical scavenging rate. (**C**) Hydroxyl radical scavenging rate. Capital and lowercase letters represent differences between VC groups and EGT groups, respectively. Data are presented as the mean values ± SD (*n* = 3). Different letters represent a significant difference (*p* < 0.05).

**Table 1 foods-13-00878-t001:** Response surface test factor level.

Factor	Number	Level		
		−1	0	1
extraction pressure (°C)	A	200	300	400
extraction time (min)	B	40	50	60
extraction solvent	C	60% EtOH	dH_2_O	80% EtOH
extraction solvent	D	1:5	1:10	1:15

**Table 2 foods-13-00878-t002:** Amino acid content of *Pleurotus citrinopileatus* Singer in different states.

Amino Acids (%)	Fresh	ND	FD	HD
Aspartic acid	9.16 ± 0.01 ^c^	10.01 ± 0.02 ^b^	12.38 ± 0.005 ^a^	9.25 ± 0.01 ^c^
Threonine	5.79 ± 0.01 ^a^	5.12 ± 0.01 ^b^	5.26 ± 0.01 ^b^	5.24 ± 0.01 ^b^
Serine	6.54 ± 0.015 ^a^	5.65 ± 0.01 ^c^	5.95 ± 0.01 ^b^	5.58 ± 0.01 ^c^
Glutamic acid	13.73 ± 0.02 ^a^	18.86 ± 0.01 ^c^	14.56 ± 0.01 ^b^	18.80 ± 0.01 ^a^
Glycine	4.44 ± 0.005 ^a^	3.94 ± 0.005 ^b^	3.75 ± 0.01 ^b^	4.16 ± 0.01 ^a^
Alanine	9.34 ± 0.005 ^a^	7.52 ± 0.01 ^b^	7.07 ± 0.015 ^c^	7.10 ± 0.01 ^c^
Cysteine	0.90 ± 0.01 ^a^	0.75 ± 0.01 ^b^	0.49 ± 0.01 ^c^	0.70 ± 0.01 ^b^
Valine	6.76 ± 0.01 ^c^	7.06 ± 0.01 ^b^	6.34 ± 0.01 ^d^	7.27 ± 0.02 ^a^
Methionine	1.12 ± 0.02 ^c^	1.99 ± 0.01 ^b^	2.24 ± 0.01 ^a^	1.93 ± 0.01 ^b^
Isoleucine	4.80 ± 0.02 ^a^	3.82 ± 0.01 ^c^	3.96 ± 0.01 ^b^	3.96 ± 0.01 ^b^
Leucine	9.17 ± 0.01 ^a^	7.88 ± 0.01 ^c^	8.77 ± 0.02 ^b^	7.99 ± 0.01 ^c^
Tyrosine	3.62 ± 0.03 ^a^	3.39 ± 0.01 ^b^	3.62 ± 0.01 ^a^	3.38 ± 0.01 ^b^
Phenylalanine	5.56 ± 0.01 ^a^	4.96 ± 0.01 ^b^	5.08 ± 0.01 ^b^	5.06 ± 0.01 ^b^
Lysine	6.38 ± 0.01 ^a^	5.70 ± 0.01 ^b^	6.32 ± 0.01 ^a^	6.35 ± 0.02 ^a^
Histidine	2.60 ± 0.01 ^a^	2.16 ± 0.01 ^b^	2.23 ± 0.005 ^b^	2.28 ± 0.005 ^b^
Arginine	3.09 ± 0.005 ^b^	5.91 ± 0.01 ^a^	5.77 ± 0.01 ^c^	5.86 ± 0.01 ^a^
Proline	6.99 ± 0.01 ^a^	5.27 ± 0.01 ^c^	6.20 ± 0.01 ^c^	5.08 ± 0.01 ^c^

Note: Lowercase letters represent significant differences among the samples (*p* < 0.05).

**Table 3 foods-13-00878-t003:** Experimental design and results of response surface method.

Run	Factor				EGT Content (mg/g)
	A Extraction Pressure	B Extraction Time	C Extraction Solvent	D Liquid–Solid Ratio	
1	0	0	0	0	4.11 ± 0.03
2	0	−1	0	1	2.97 ± 0.02
3	1	0	0	−1	2.73 ± 0.01
4	0	1	0	1	2.54 ± 0.01
5	−1	0	−1	0	2.97 ± 0.02
6	−1	1	0	0	3.59 ± 0.03
7	0	−1	0	−1	2.81 ± 0.02
8	−1	−1	0	0	2.75 ± 0.01
9	1	1	0	0	1.23 ± 0.01
10	0	1	1	0	1.55 ± 0.01
11	0	−1	−1	0	2.15 ± 0.02
12	0	−1	1	0	2.74 ± 0.04
13	1	0	−1	0	2.39 ± 0.02
14	−1	0	0	−1	3.41 ± 0.04
15	1	−1	0	0	2.76 ± 0.02
16	−1	0	1	0	3.11 ± 0.02
17	0	0	0	0	4.09 ± 0.04
18	0	0	−1	−1	2.54 ± 0.02
19	0	0	1	−1	2.85 ± 0.03
20	0	1	0	−1	2.43 ± 0.02
21	0	0	0	0	4.17 ± 0.02
22	1	0	1	0	1.85 ± 0.01
23	0	1	−1	0	2.29 ± 0.01
24	0	0	−1	1	2.66 ± 0.02
25	1	0	0	1	2.46 ± 0.05
26	−1	0	0	1	3.71 ± 0.03
27	0	0	1	1	2.55 ± 0.01

**Table 4 foods-13-00878-t004:** ANOVA of the second-order polynomial model for the yield of EGT.

Source	SS ^a^	DF ^b^	MS ^c^	*F*-Value	*p*-Value
Model	13.37	14	0.9550	88.53	<0.0001 **
A	3.12	1	3.12	289.35	<0.0001 **
B	0.5419	1	0.5419	50.23	<0.0001 **
C	0.0102	1	0.0102	0.9464	0.3498 NS
D	0.0012	1	0.0012	0.1112	0.7445 NS
AB	1.40	1	1.40	130.18	<0.0001 **
AC	0.1156	1	0.1156	10.72	0.0067 *
AD	0.0812	1	0.0812	7.53	0.0178 *
BC	0.4422	1	0.4422	41.00	<0.0001 **
BD	0.0006	1	0.0006	0.0579	0.8138 NS
CD	0.0441	1	0.0441	4.09	0.0661 NS
A²	1.72	1	1.72	159.70	<0.0001 **
B²	4.94	1	4.94	457.65	<0.0001 **
C²	5.14	1	5.14	476.87	<0.0001 **
D²	1.23	1	1.23	114.31	<0.0001 **
Residual	0.1294	12	0.0108		
Lack of fit	0.1260	10	0.0126	7.27	0.1269 NS
Pure error	0.0035	2	0.0017		
Cor. total	13.50	26			
R^2^	0.9904				
R^2^Adj	0.9792				
R^2^pred	0.9457				

SS ^a^, Sum of squares. DF ^b^, Degree of freedom. MS ^c^, Mean square deviation. NS, No significance. * Significant difference (*p* < 0.05). ** Highly significant difference (*p* < 0.01).

**Table 5 foods-13-00878-t005:** Content change of each component in the sample.

Sample	Total Sugar (%)	Protein (%)	EGT Purity (%)
Extracted solution	59.89 ± 2.67%	6.48 ± 1.14%	0.42 ± 0.32%
Filtered solution	15.37 ± 1.31%	1.24 ± 0.52%	32.87 ± 2.62%

## Data Availability

The original contributions presented in the study are included in the article, further inquiries can be directed to the corresponding authors.

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
