# Peer review of "Effects of Drying Process and High Hydrostatic Pressure on Extraction of Antioxidant Ergothioneine from Pleurotus citrinopileatus Singer"

_foods, 2024, doi:10.3390/foods13060878_

Round 1

Reviewer 1 Report

Comments and Suggestions for Authors

A very interesting paper. I would recommend the following suggestions for further improvement:

1. Include the chemical structures of EGT and thioketone to understand the relationship between the two.

2.      Include a photo or drawing of Pleurotus citrinilileatus Singer mushroom.

3.      I do not agree with the statement “High hydrostatic pressure (HHP) is a novel technique with many benefits, including low energy consumption”

4.      Lines 86-88. “The pre-frozen mushrooms were rapidly spread on the dry layer, and the cold depression temperature was set at -70 °C, with a vacuum pressure of 200 Pa for vacuum drying, until freeze-drying reaches a constant weight.” This statement is confusing. Vacuum and freeze-drying are two distinct drying processes. For Freeze drying, the temperature and pressure should be brought to below the eutectic point of the product and the temperature is gradually increased to effect sublimation. Whereas vacuum drying is evaporation at low temperature and pressure.

5.      Lines 99-101. “…the samples were cooled, transferred to a 10 mL volume bottle and fixed volume using 1:1 hydrochloric acid solution. 0.25 mL liquid nitrogen sample was air-dried, and then 0.02 mol/L hydrochloric acid solution was fixed volume to 10 mL.”  Use the phrase ‘made up to volume’ instead of fixed volume.

6.      Line 100. “0.25 mL liquid nitrogen sample was air-dried….” Never start a sentence with a number. You can start the sentence by adding the phrase “An aliquot of 0.25 mL……”

7.      Line 105. The sample was cut into 2 mm thick slices… what sample?

8.      Line 107. “… and freeze-dried for 5 h under vacuum”. Is this freeze-dried or vacuum-dried? If freeze dried delete the words ‘under vacuum’. Freeze drying is always done under vacuum.

9.      Line 111. “The TEM was used to observe the tissue (5 mm×5 mm×1 mm) cut from the lid of the…” Are you talking about the cap of the mushroom?

10.  Line 119. “….finally observed and photographed by TEM at 5000x and 10,000x”. add the word magnification after 10,000x

11.  Line 124. “…and the volume was fixed at 100 ml with 60% ethanol”. Use the phrase made up to 100 mL instead of fixed at 100 mL

12.  Equation 1 is incorrect. If you are using fresh mushrooms, they will have a certain moisture content which will give false results when using Eq 1.

13.  Lines 149-150. “The crude extracts were diluted to 20 mg/mL with EGT and injected into the column….” This is incorrect. The crude extract was diluted to 20 mg/mL of EGT using what?

14.  Lines 151-154. The sentence is incomprehensible.

15.  Sample preparation for LC-MS is not quite clear.

16.  Figure 1 is not cited in the text.

17.  Line 206. EGT content of 2.63±0.06 mg/g in fresh mushroom….. Was the moisture content accounted for in the calculation of EGT in the fresh mushroom? It was not shown in Equation 1

18.  Lines 206-207. “The fresh mushrooms (2.63±0.06B mg/g, P < 0.05) and freeze vacuum drying (FVD) mushrooms (2.53±0.04B mg/g) had higher EGT yield”. This is not true. You found that natural air drying gave the highest EGT value.

19.  Lines 204 -211. Incomprehensible sentences. When you say EGT contents of fresh and FVD samples were higher, you have not stated what they are higher than.

20.  What is the possible reason for the discrepancy in the values of EGT found in mushrooms dried by different methods?

21.  I suppose the letters in superscript (you should indicate below the table what the superscripts mean) following values of Cysteine & Histidine in Table 2 denote the significance values between the samples. If so, I do not believe them. For example, the values for Histidine are given below. It shows values for Fresh and FVD have no significant differences, but there is a significant difference between FVD and HD, which is hard to believe.

Histidine

2.60±0.005c

2.16±0.01b

2.23±0.005c

2.28±0.005a

22.  Therefore, your rationale for the difference in EGT content in the different samples is incorrect.

23.  Lines 350-359. Figure 7 should appear before Figure 8 in the text.

24.  Line 400-402. Conclusion “The present study demonstrated that different drying methods caused different degrees of contraction and destruction of the tissue structure of the mushroom and changed the cellular material of the mushroom, which affected the anabolism of amino acids”.

This is pure speculation and conjecture. Your rationale that the Cysteine and Histidine contents were altered by the different drying techniques was not true. Table 2 does show otherwise. 

Comments on the Quality of English Language

There were many syntax errors in the use of the English language and the construction of sentences. They need to be vastly improved before the paper can be accepted.

My recommendation is to reject the paper until all the corrections are made.

Author Response

Itemized list of changes addressing reviewer comments

Thank you very much for taking the time to review this manuscript. Please find the detailed responses below and the corrections highlighted in the re-submitted files.

Reviewer #1:

A very interesting paper. I would recommend the following suggestions for further improvement:

Comments 1: Include the chemical structures of EGT and thioketone to understand the relationship between the two.

Response 1: Thank you for pointing this out. We agree with this comment. Therefore, we added the structure diagram of EGT and illustrated on page 1, lines 34-35. [Fig. 1 (A) shows that the compound EGT undergoes a tautomeric equilibrium between its thiol and thione forms.

Fig. 1 (A) Chemical structure of ergothioneine (EGT) thione-thiol tautomers.]

Comments 2:  Include a photo or drawing of Pleurotus citrinilileatus Singer mushroom.

Response 2: Thank you for pointing this out. We agree with this comment. Therefore, we added the photo of Pleurotus citrinilileatus Singer and illustrated on page 1, line 44. [ Pleurotus citrinopileatus Singer (Fig.1B) is rich in EGT

Fig. 1 (B) Pho-tograph of Pleurotus citrinopileatus Singer mushrooms.]

Comments 3: I do not agree with the statement “High hydrostatic pressure (HHP) is a novel technique with many benefits, including low energy consumption”

Response 3: Thank you for pointing this out. We agree with this comment. Therefore, we have revised the writing of this paragraph to focus on the application of HHPE. Lines 52-60 [High hydrostatic pressure extraction (HHPE) involves applying 100-1000 MPa hydrostatic pressure to the material and solvent under normal temperature conditions, holding the pressure for a period, and reducing the pressure to normal pressure in a short time after holding the pressure. A large pressure difference between inside and outside the cell will quickly transfer the active ingredients in the material to the ex-traction agent, thus completing the high-pressure extraction process [18]. Recently, high hydrostatic pressure (HHP) has found widespread application in the extraction of naturally active substances from mushrooms and other plants. For example, natural melanin has been extracted from the wild mushroom Auricularia auricula [19], bioactive peptides have been extracted from mushrooms [20], and phenolic compounds have been extracted from various plants [18].]

Comments 4: Lines 86-88. “The pre-frozen mushrooms were rapidly spread on the dry layer, and the cold depression temperature was set at -70 °C, with a vacuum pressure of 200 Pa for vacuum drying, until freeze-drying reaches a constant weight.” This statement is confusing. Vacuum and freeze-drying are two distinct drying processes. For Freeze drying, the temperature and pressure should be brought to below the eutectic point of the product and the temperature is gradually increased to effect sublimation. Whereas vacuum drying is evaporation at low temperature and pressure.

Response 4: Thank you for pointing this out. We agree with this comment. Freeze vacuum drying (FVD) has been modified to Freeze drying (FD), and the whole article has been modified. In addition, the device model has been added. Lines 99-104. [Freeze drying (FD): Fresh P. citrinopileatus was subjected to pre-freezing at -20 °C for 3 h and then placed in the drying room of the freeze-drying machine (Pilot7-12E, BIOCOOL Instrument Co., Ltd, Beijing, China) for freeze-drying for 24 h, until the samples reached a constant weight. The FD mushrooms were then broken up using a grinder (IKA MuItiDrive BASIC S025, IKA (Guangzhou) Instrument Equipment Co., Ltd, China) and sifted through a 50-mesh screen. The mushroom powder was sealed and stored in a cool and dry place for further use.]

Comments 5:  Lines 99-101. “…the samples were cooled, transferred to a 10 mL volume bottle and fixed volume using 1:1 hydrochloric acid solution. 0.25 mL liquid nitrogen sample was air-dried, and then 0.02 mol/L hydrochloric acid solution was fixed volume to 10 mL.”  Use the phrase ‘made up to volume’ instead of fixed volume.

Response 5: Thank you for pointing this out. We agree with this comment. According to your comments, “fixed volume” has been replaced by “made up to volume” in the manuscript. Line 118. [and made up to a final volume of 10 mL with 0.02 mol/L HCL.]

Comments 6: Line 100. “0.25 mL liquid nitrogen sample was air-dried….” Never start a sentence with a number. You can start the sentence by adding the phrase “An aliquot of 0.25 mL……”

Response 6: Thank you for pointing this out. We agree with this comment. According to your comments, we rewrote the passage. Lines 156-160. [Following hydrolysis, the samples were cooled, transferred to a 10 mL volume bottle and made up to a final volume of 10 mL with 0.02 mol/L HCL. The sample solution was filtered through a 0.22 μm membrane. Then, an aliquot of 20 μL of the mixed amino acid standard working solution and sample determination solution was injected into the amino acid analyzer (LA8080, HITACHI Co., Ltd, Japan).]

Comments 7: Line 105. The sample was cut into 2 mm thick slices… what sample?

Response 7: Thank you for pointing this out. The samples were the mushrooms in different states. We express it in detail in the manuscript. Line 124. [The mushrooms in different states were cut into 2 mm thick slices with a blade….]

Comments 8:  Line 107. “… and freeze-dried for 5 h under vacuum”. Is this freeze-dried or vacuum-dried? If freeze dried delete the words ‘under vacuum’. Freeze drying is always done under vacuum.

Response 8: Thank you for pointing this out. We agree with this comment. It is freeze-dried and we delete the words ‘under vacuum’. Lines 126-127. [and the samples were freeze-dried for 5 h.]

Comments 9:  Line 111. “The TEM was used to observe the tissue (5 mm×5 mm×1 mm) cut from the lid of the…” Are you talking about the cap of the mushroom?

Response 9: Thank you for pointing this out. Yes. We are talking about the cap of the mushroom. TEM was used to observe the cap of mushrooms. We added a specific description. Lines 130-131. [A TEM was used to observe the tissue (5 mm × 5 mm × 1 mm) cut from the cap of the different states of mushrooms (fresh, ND, FD, and HD).]

Comments 10: Line 119. “….finally observed and photographed by TEM at 5000x and 10,000x”. add the word magnification after 10,000x

Response 10: Thank you for pointing this out. We agree with this comment. According to your comments, we added word magnification after 10,000x. Line 139. [….by TEM at 5000× and 10,000× magnification.]

Comments11: Line 124. “…and the volume was fixed at 100 ml with 60% ethanol”. Use the phrase made up to 100 mL instead of fixed at 100 mL.

Response11: Thank you for pointing this out. We agree with this comment. According to your comments, “fixed volume” has been replaced by “made up to volume” in the manuscript. Line 148. [and the volume was made up to 100 mL]

Comments 12:  Equation 1 is incorrect. If you are using fresh mushrooms, they will have a certain moisture content which will give false results when using Eq 1.

Response 12: Thank you for pointing this out. We agree with this comment. Therefore, we modified the Eq 1 and added the water content factor of mushrooms into the calculation of the formula, and finally the EGT content was dry weight, avoiding the error caused by fresh mushrooms and dried mushrooms. Line 154. [, where C is the EGT content of EGT by HPLC, V is the constant volume, m is the weight of the mushrooms and w is the moisture content of the mushroom.]

Comments 13: Lines 149-150. “The crude extracts were diluted to 20 mg/mL with EGT and injected into the column….” This is incorrect. The crude extract was diluted to 20 mg/mL of EGT using what?

Response 13: Thank you for pointing this out. We agree with this comment. We rewrote the passage. Lines 186-188. [The concentrated liquid was diluted to 20 mg/mL with dH2O and injected into the column using a digital display pump at a constant flow rate of 2 BV/h.]

Comments 14: Lines 151-154. The sentence is incomprehensible.

Response 14: Thank you for pointing this out. We agree with this comment. We rewrote the passage. Lines188-191. [After the resin adsorption saturation, the sample was eluted using 20 times the volume of dH2O to obtain eluent. The eluent was collected and freeze-dried to obtain a freeze-dried powder for quantitative and qualitative analysis using UPLC-Q-TOF-MSE (Q Exactive Plus, Thermo Fisher Scientific Inc., USA)]

Comments 15: Sample preparation for LC-MS is not quite clear.

Response 15: Thank you for pointing this out. We agree with this comment. We rewrote the passage. Lines 197-205. [Detection conditions: Aqueous 0.1% formic acid (A) and acetonitrile (B) were used as the mobile phase. Gradient elution was performed as follows: 0–1.5 min, 5% B; 1.6–3.5 min, 10% B; 3.6–5.0 min, 5%. The loading quantity and speed were 2 μL and 0.3 mL/ min, respectively. Using an ESI source, EGT were identified by MS/MS. Positive ionization mode electrospray source with a range of 50–255 m/z was used for MS. EGT was iden-tified based on the ionic group, retention time, and peak area. The EGT standard was used as an external standard, and the EGT was quantified according to the standard curve y = 320.287 x +359.74 (R2 = 0.9997) in Fig. 2, where y represents the response and x represents the EGT concentration (ng/mL).]

Comments 16: Figure 1 is not cited in the text.

Response 16: Thank you for pointing this out. We agree with this comment. Based on the previous changes, Fig.1 becomes Fig.2 here, and we have added a cite to the text. Lines 203-204, [and the EGT was quantified according to the standard curve y = 320.287 x +359.74 (R2 = 0.9997) in Fig. 2,]

Comments 17:  Line 206. EGT content of 2.63±0.06 mg/g in fresh mushroom….. Was the moisture content accounted for in the calculation of EGT in the fresh mushroom? It was not shown in Equation 1

Response 17: Thank you for pointing this out. It is the moisture content accounted for in the calculation of EGT in the fresh mushroom. Therefore, we modified the Eq 1 and added the water content factor of mushrooms into the calculation of the formula, and finally the EGT content was dry weight, avoiding the error caused by fresh mushrooms and dried mushrooms. Line 154. [, where C is the EGT content of EGT by HPLC, V is the constant volume, m is the weight of the mushrooms and w is the moisture content of the mushroom.]

Comments 18: Lines 206-207. “The fresh mushrooms (2.63±0.06B mg/g, P < 0.05) and freeze vacuum drying (FVD) mushrooms (2.53±0.04B mg/g) had higher EGT yield”. This is not true. You found that natural air drying gave the highest EGT value.

Response 18: Thank you for pointing this out. We agree with this comment. We have revised the expression in the article. Lines 246-250. [Fig. 3 shows that the yield of EGT was the highest in the ND-treated mushrooms (3.72 ± 0.06A mg/g d.w., P < 0.05). The EGT yields of fresh mushrooms (2.63 ± 0.06B mg/g d.w.) and FD-treated mushrooms (2.53 ± 0.04B mg/g d.w.) were lower than those of ND-treated mushrooms. The yield of EGT in HD-treated mushrooms was the lowest (2.37 ± 0.07C mg/g d.w.).]

Comments 19: Lines 204 -211. Incomprehensible sentences. When you say EGT contents of fresh and FVD samples were higher, you have not stated what they are higher than.

Response 19: Thank you for pointing this out. We have revised the expression in the article. Lines 246-250. [Fig. 3 shows that the yield of EGT was the highest in the ND-treated mushrooms (3.72 ± 0.06A mg/g d.w., P < 0.05). The EGT yields of fresh mushrooms (2.63 ± 0.06B mg/g d.w.) and FD-treated mushrooms (2.53 ± 0.04B mg/g d.w.) were lower than those of ND-treated mushrooms. The yield of EGT in HD-treated mushrooms was the lowest (2.37 ± 0.07C mg/g d.w.).]

Comments 20: What is the possible reason for the discrepancy in the values of EGT found in mushrooms dried by different methods?

Response 20: Thank you for pointing this out. We have included the reasons. Lines 265-273. [Therefore, we speculate that one of the reasons for the high EGT content under ND conditions is that natural drying takes the longest time, and the water activity is high most of the time (for approximately the first 10 h, the water activity is higher than 50%). Thus, the key synthetases can remain active for a long time and can make sufficient use of oxygen to catalyze the precursor amino acid (histidine) to synthesize EGT, resulting in a reduction in the histidine content. In addition, in the freeze-drying process, the activity of the enzyme is inhibited, and there is a lack of oxygen. Thus, the degree of synthesis is no different from that in the fresh state, so the EGT content is not significantly different.]

Comments 21:  I suppose the letters in superscript (you should indicate below the table what the superscripts mean) following values of Cysteine & Histidine in Table 2 denote the significance values between the samples. If so, I do not believe them. For example, the values for Histidine are given below. It shows values for Fresh and FVD have no significant differences, but there is a significant difference between FVD and HD, which is hard to believe.

Histidine 2.60±0.005c    2.16±0.01b     2.23±0.005c    2.28±0.005a

Histidine

2.60±0.01a

2.16±0.01b

2.23±0.005b

2.28±0.005b

Response 21: Thank you for pointing this out. We agree with this comment. According to your comments, we indicate the letters in superscript following values of Cysteine & Histidine in Table 2 denote the significance values between the samples. Line 302. [Note: Lowercase letters represent significant differences among the samples (P < 0.05).] In addition, when we examined the manuscript, we found that the letter identification was incorrect, so we made a correction. For example, the values for Histidine are given below.

Moreover, the results are analyzed again in this text. Lines 256-273. [As shown in Table 2, the histidine contents of the mushrooms in different states (Fresh, HD, FD, and ND) were 2.60 ± 0.01a, 2.28 ± 0.005b, 2.23 ± 0.005b, and 2.16 ± 0.01b, respectively. The contents showed downward trend and were negatively correlated with EGT content. The fungal synthesis pathway is shown in Fig. 4. Studies have shown that the fungal L-EGT synthesis pathway involves the transfer of three methyl groups from S-adenosylmethionine (SAM) to histidine to form histidine trimethylenolide by the synthetase Egt1. Oxygen and cysteine are then used to catalyze histidine trimethylenolide to form hep-tylenocysteine sulfoxide, and finally, L-EGT is synthesized by the Egt2 synthetase enzyme [30,31]. Histidine and oxygen are the key components of this pathway [32,33]. Therefore, we speculate that one of the reasons for the high EGT content under ND conditions is that natural drying takes the longest time, and the water activity is high most of the time (for approximately the first 10 h, the water activity is higher than 50%). Thus, the key synthetases can remain active for a long time and can make sufficient use of oxygen to catalyze the precursor amino acid (histidine) to synthesize EGT, resulting in a reduction in the histidine content. In addition, in the freeze-drying process, the ac-tivity of the enzyme is inhibited, and there is a lack of oxygen. Thus, the degree of synthesis is no different from that in the fresh state, so the EGT content is not significantly different.]

Comments 22: Therefore, your rationale for the difference in EGT content in the different samples is incorrect.

Response 22: Thank you for pointing this out. We agree with this comment. According to the revision of the amino acid results, we also revised the analysis of the results. Lines 256-273. [As shown in Table 2, the histidine contents of the mushrooms in different states (Fresh, HD, FD, and ND) were 2.60 ± 0.01a, 2.28 ± 0.005b, 2.23 ± 0.005b, and 2.16 ± 0.01b, respectively. The contents showed downward trend and were negatively correlated with EGT content. The fungal synthesis pathway is shown in Fig. 4. Studies have shown that the fungal L-EGT synthesis pathway involves the transfer of three methyl groups from S-adenosylmethionine (SAM) to histidine to form histidine trimethylenolide by the synthetase Egt1. Oxygen and cysteine are then used to catalyze histidine trimethylenolide to form hep-tylenocysteine sulfoxide, and finally, L-EGT is synthesized by the Egt2 synthetase enzyme [30,31]. Histidine and oxygen are the key components of this pathway [32,33]. Therefore, we speculate that one of the reasons for the high EGT content under ND conditions is that natural drying takes the longest time, and the water activity is high most of the time (for approximately the first 10 h, the water activity is higher than 50%). Thus, the key synthetases can remain active for a long time and can make sufficient use of oxygen to catalyze the precursor amino acid (histidine) to synthesize EGT, resulting in a reduction in the histidine content. In addition, in the freeze-drying process, the ac-tivity of the enzyme is inhibited, and there is a lack of oxygen. Thus, the degree of synthesis is no different from that in the fresh state, so the EGT content is not significantly different.]

Comments 23: Lines 350-359. Figure 7 should appear before Figure 8 in the text.

Response 23: Thank you for pointing this out. We agree with this comment. According to your comments, we have added a cite to the text. Lines 409-410. [Fig. 9 shows the acquisition time and the mass-to-charge ratio of the standard.]

Comments 24: Line 400-402. Conclusion “The present study demonstrated that different drying methods caused different degrees of contraction and destruction of the tissue structure of the mushroom and changed the cellular material of the mushroom, which affected the anabolism of amino acids”.

This is pure speculation and conjecture. Your rationale that the Cysteine and Histidine contents were altered by the different drying techniques was not true. Table 2 does show otherwise. 

Response 24: Thank you for pointing this out. We agree with this comment. According to the previous changes, we rewrote the conclusion. Lines 462-469. [We demonstrated that different drying methods decreased the amino acid content in mushrooms and at the same time caused different degrees of contraction and damage to the tissue structure of mushroom. In the experimental results, under different treatment conditions, the decreasing trend of histidine as a precursor substance was negatively correlated with the increasing trend of EGT content, and the content of EGT was the highest under ND treatment with sufficient oxygen. Combined with the syn-thetic pathway of EGT in fungi (Fig. 4), we speculated that ND conditions were condu-cive to the synthesis of EGT in fungi.]

Response to Comments on the Quality of English Language

Point 1: There were many syntax errors in the use of the English language and the construction of sentences. They need to be vastly improved before the paper can be accepted.

Response 1: Thank you for pointing this out. We agree with this comment. According to your comments, we used a professional editing service to polish our revised manuscript.

Reviewer 2 Report

Comments and Suggestions for Authors

The work and research results are interesting. However, I notice some inconsistency. The work determined the optimal extraction conditions, but the extraction process was assessed for completely different conditions. Four materials were also used: fresh and 3 types of dried. The analysis omitted the fact that the type of material may have a significant impact on both the extraction efficiency and the characteristics of the extract properties. It is also not stated for which material the purification process and the assessment of antioxidant properties were carried out. It is unclear why this was not done for the extract obtained under the conditions resulting from process optimization

detailed remarks

Line 85 Please provide the type of freeze dryer

Line 91 Please provide the type of dryer and drying conditions: natural or forced convection?

Lina 118 Necessary spaces (30,50,70,80,95%),

Line 123 specify the rotational acceleration (g), not rpm. If rpm, you must also provide the centrifuge rotor diameter

Line 121-3 Why was extraction performed under these parameters? Optimal parameters have been experimentally determined and maybe these should be used?

Formula 1 and line 127-8, the method of calculation and the unit are unclear. Does dw mean dry weight? If so, the formula should take into account the water content in the mushrooms. If we do not take into account the water content, the result is mg / g of fresh or dried mushroom

Line 133 should be MPa

Subsection 2.4 What material was it based on? dried or fresh? Why was the type of material omitted as a factor also affecting extraction

Line 154, 157 space required10 mL 0.22 μm

Line 167 water.The

Section 3.1 Please try to explain the higher content of EGT in the NC material. Please provide the duration of each type of drying - maybe there will be an explanation here?

Line 223 It is worth referring to the optimal conditions for this enzyme. Maybe it's a matter of how active this enzyme is. It is worth remembering that drying in natural conditions is a long-term process with relatively high water activity for a large part of the drying time, hence the mass of the products of the reaction catalyzed by a given enzyme is higher. During freeze-drying, enzyme activity is inhibited by freezing and the lack of water as a reaction medium. However, during HD drying, the time during which the water activity in the mushroom is high is much shorter compared to ND.

Line 234 "and causing nutrient loss" - this does not apply to freeze-dried products, this happens when freezing and then thawing accompanied by leakageFig 2. signature under the drawing, not on the next page

Line 267-9 t is worth referring to the equation of unsteady mass transport (Fick's second law). A large pressure difference is also the driving force for mass movement (movement of the extracted substance), and the amount of extracted substance depends on time

subsection 3.4. and 3.5. Please explain which of the extracts was purified. It seems that this one was obtained using the traditional method. Why was the extract obtained under optimal conditions not tested?

Line 417-8  „provides a new idea for the synthesis of EGT 417 in fungus, so that it can be used in food applications.” It is necessary to specify what this new idea is. I don't know what the authors had in mind, so I can't confirm that this statement is justified by the results obtained

Author Response

Itemized list of changes addressing reviewer comments

Thank you very much for taking the time to review this manuscript. Please find the detailed responses below and the corrections highlighted in the re-submitted files.

The work and research results are interesting. However, I notice some inconsistency. The work determined the optimal extraction conditions, but the extraction process was assessed for completely different conditions. Four materials were also used: fresh and 3 types of dried. The analysis omitted the fact that the type of material may have a significant impact on both the extraction efficiency and the characteristics of the extract properties. It is also not stated for which material the purification process and the assessment of antioxidant properties were carried out. It is unclear why this was not done for the extract obtained under the conditions resulting from process optimization

Comments 1: Line 85 Please provide the type of freeze dryer.

Response 1: Thank you for pointing this out. We agree with this comment. According to your comments, we provide the type of freeze dryer. Lines 100-101. […. then placed in the drying room of the freeze-drying machine (Pilot7-12E, BIOCOOL Instrument Co., Ltd, Beijing, China) for freeze-drying for 24 h….]

Comments 2: Line 91 Please provide the type of dryer and drying conditions: natural or forced convection?

Response 2: Thank you for pointing this out. We agree with this comment. According to your comments, we provide the type of dryer and drying conditions, it is natural convection. Lines 106-107. [Fresh P. citrinopileatus were kept placed in a natural convection drying oven (DHG-9015A, Yiheng Scientific Instrument Co., Ltd, Shanghai, China) at….]

Comments 3: Lina 118 Necessary spaces (30,50,70,80,95%).

Response 3: Thank you for pointing this out. We agree with this comment. According to your comments, we made revisions to the text. Lines 137-138. [(30%, 50%, 70%, 80%, and 95%),]

Comments 4: Line 123 specify the rotational acceleration (g), not rpm. If rpm, you must also provide the centrifuge rotor diameter.

Response 4: Thank you for pointing this out. We agree with this comment. According to your comments, we made revisions to the text. Lines 147-148. [….and centrifuged at 9500 ×g for 10 min.]

Comments 5: Line 121-3 Why was extraction performed under these parameters? Optimal parameters have been experimentally determined and maybe these should be used?

Response 5: Thank you for pointing this out. This is the part where we extract EGT in the existing way. Because this part is the first part of the research content, to explore the difference of EGT in mushrooms obtained by different drying methods under the traditional extraction method. The aim is to select the raw material with the highest EGT content (which state of mushroom) according to the existing extraction methods. After selecting the mushrooms with the highest content, on this basis, a method can be optimized to improve the extraction rate. Optimization method is the research content of the second part, so it is not used in this part.

Comments 6: Formula 1 and line 127-8, the method of calculation and the unit are unclear. Does dw mean dry weight? If so, the formula should take into account the water content in the mushrooms. If we do not take into account the water content, the result is mg / g of fresh or dried mushroom.

Response 6: Thank you for pointing this out. It is the moisture content accounted for in the calculation of EGT in the fresh mushroom. Therefore, we modified the Eq 1 and added the water content factor of mushrooms into the calculation of the formula, and finally the EGT content was dry weight, avoiding the error caused by fresh mushrooms and dried mushrooms. Line 154. [, where C is the EGT content of EGT by HPLC, V is the constant volume, m is the weight of the mushrooms and w is the moisture content of the mushroom.]

Comments 7: Line 133 should be MPa.

Response 7: Thank you for pointing this out. We agree with this comment. According to your comments, “Mpa” as been replaced by “MPa”. Line 165. [400 MPa]

Comments 8: Subsection 2.4 What material was it based on? dried or fresh? Why was the type of material omitted as a factor also affecting extraction.

Response 8: Thank you for pointing this out. The material is dried (ND mushrooms). We have added a specific description in the article. Line 170. [The ND mushrooms were used as raw materials.] Because we are divided into two research parts, the first part is to explore the influence of different pretreatment on the mushroom tissue and amino acid content, and thus the influence on the EGT content, and select an optimal raw material. On this basis, the extraction process was optimized. The type of material was treated as a separate part of the study and therefore not considered as a factor in optimization.

Comments 9: Line 154, 157 space required10 mL 0.22 μm.

Response 9: Thank you for pointing this out. We agree with this comment. According to your comments, we made revisions to the text. Lines 194 and 196. [10 mL, 0.22 μm]

Comments 10: Line 167 water. The Section 3.1 Please try to explain the higher content of EGT in the NC material. Please provide the duration of each type of drying - maybe there will be an explanation here?

Response 10: Thank you for pointing this out. We agree with this comment. According to your comments, we provide the duration of each type of drying in Section the 2.2.1. Lines 94, 101, and 108. [and allowed to dry naturally for 72 h until it reached a constant weight; for freeze-drying for 24 h; at 60 °C and dried for 24 h until a constant weight was achieved.] At the same time, we explain the effect of drying time in the Section 3.1 result analysis. Lines 265-270. [Therefore, we speculate that one of the reasons for the high EGT content under ND conditions is that natural drying takes the longest time, and the water activity is high most of the time (for approximately the first 10 h, the water activity is higher than 50%). Thus, the key synthetases can remain active for a long time and can make sufficient use of oxygen to catalyze the precursor amino acid (histidine) to synthesize EGT, resulting in a reduction in the histidine content.]

Comments11: Line 223 It is worth referring to the optimal conditions for this enzyme. Maybe it's a matter of how active this enzyme is. It is worth remembering that drying in natural conditions is a long-term process with relatively high water activity for a large part of the drying time, hence the mass of the products of the reaction catalyzed by a given enzyme is higher. During freeze-drying, enzyme activity is inhibited by freezing and the lack of water as a reaction medium. However, during HD drying, the time during which the water activity in the mushroom is high is much shorter compared to ND.

Response11: Thank you for pointing this out. We agree with this comment. According to your comments, we have enriched the results analysis in this section. Lines 265-273. [Therefore, we speculate that one of the reasons for the high EGT content under ND conditions is that natural drying takes the longest time, and the water activity is high most of the time (for approximately the first 10 h, the water activity is higher than 50%). Thus, the key synthetases can remain active for a long time and can make sufficient use of oxygen to catalyze the precursor amino acid (histidine) to synthesize EGT, resulting in a reduction in the histidine content. In addition, in the freeze-drying process, the ac-tivity of the enzyme is inhibited, and there is a lack of oxygen. Thus, the degree of synthesis is no different from that in the fresh state, so the EGT content is not significantly different.]

Comments 12: Line 234 "and causing nutrient loss" - this does not apply to freeze-dried products, this happens when freezing and then thawing accompanied by leakage. Fig 2. signature under the drawing, not on the next page.

Response 12: Thank you for pointing this out. We agree with this comment. According to your comments, we deleted "and causing nutrient loss". Line 280. And we fixed the formatting of Fig. 3 (Original Fig. 2) Line 296.

Comments 13: Line 267-9 t is worth referring to the equation of unsteady mass transport (Fick's second law). A large pressure difference is also the driving force for mass movement (movement of the extracted substance), and the amount of extracted substance depends on time.

Response 13: Thank you for pointing this out. We agree with this comment. According to your comments, we included a reference to this law. Lines 320-323. [According to Fick's second law [40], the pressure difference can be used as the force of the extracted substance, and the greater the pressure difference, the stronger the force, the more extract can be obtained. At the same time, the quality of the extracted material is also related to the extraction time.]

Comments 14: subsection 3.4. and 3.5. Please explain which of the extracts was purified. It seems that this one was obtained using the traditional method. Why was the extract obtained under optimal conditions not tested?

Response 14: Thank you for pointing this out. It is the extract obtained under optimal conditions. According to your comments, we have added a detailed statement in the article. Lines 401-402, 429-430. [According to the optimized extraction conditions, the HHPE crude extract was obtained from ND mushroom powder. The crude extract was separated and purified; The antioxidant activity of EGT purified product obtained in Section 3.4 was determined.]

Comments 15: Line 417-8, provides a new idea for the synthesis of EGT 417 in fungus, so that it can be used in food applications.” It is necessary to specify what this new idea is. I don't know what the authors had in mind, so I can't confirm that this statement is justified by the results obtained.

Response 15: Thank you for pointing this out. The new idea is we speculated that ND conditions were conducive to the synthesis of EGT in fungi. This part of the content, we have redescribed. Lines 462-469. [We demonstrated that different drying methods decreased the amino acid content in mushrooms and at the same time caused different degrees of contraction and damage to the tissue structure of mushroom. In the experimental results, under different treatment conditions, the decreasing trend of histidine as a precursor substance was negatively correlated with the increasing trend of EGT content, and the content of EGT was the highest under ND treatment with sufficient oxygen. Combined with the syn-thetic pathway of EGT in fungi (Fig. 4), we speculated that ND conditions were conducive to the synthesis of EGT in fungi.]

Reviewer 3 Report

Comments and Suggestions for Authors

See attached file.

Author Response

Itemized list of changes addressing reviewer comments

Thank you very much for taking the time to review this manuscript. Please find the detailed responses below and the corrections highlighted in the re-submitted files.

I am very grateful for the invitation on to review the manuscript foods-2883365 by Changge Zhang and co-authors “Extraction on of antioxidant ergothioneine from Pleurotus citrinilileatus Singer using high hydrostatic pressure”. The study aimed to demonstrate how different drying methods (natural ventilation on drying, freeze vacuum drying, hot air drying) affect the properties of Pleurotus citrinilileatus Singer and its EGT content, to optimize the extraction process, to purify the crude extract, and to study the antioxidant activity of different concentrations of EGT.

The work is interesting but needs adjustments to increase the quality of the material.

Abstract

Comments 1: Abstract is 260 words, exceeds journals’ requirement of 250 words.

Response 1: Thank you for pointing this out. We agree with this comment. According to your comments, we have condensed the summary. Lines 10-27. [This study evaluated the effects of different drying techniques on the physicochemical properties of Pleurotus citrinopileatus Singer (P. citrinopileatus), focusing on the ergothioneine (EGT) contents. The P. citrinopileatus was subjected to natural ventilation drying (ND), freeze-drying (FD), and hot-air drying (HD). EGT was extracted using high hydrostatic pressure extraction (HHPE), and response surface methodology (RSM) was employed with four variables to optimize the extraction parameters. The crude EGT extract was purified by ultrafiltration and anion resin purification, and its antioxidant activity was investigated. The results showed that the ND method effectively disrupted mushroom tissues, promoting amino acid anabolism, thereby increasing the EGT content in mushrooms. Based on RSM, the optimum extracting conditions were a pressure of 250 MPa, extraction time of 52 min, distilled water (dH2O) as the extraction solvent, and 1:10 a liquid–solid ratio, which yielded the highest EGT content of 4.03 ± 0.01 mg/g d.w. UPLC-Q-TOF-MSE was performed to assess the purity of the samples (purity: 86.34±3.52%), and MS2 information of the main peak showed primary ions (m/z 230.1) and secondary cations (m/z 186.1050, m/z 127.0323) consistent with standard products. In addition, compared with ascorbic acid (VC), EGT showed strong free radical scavenging ability, especially for hydroxyl and ATBS radicals, at more than 5 mmol/L. These findings indicate that the extraction and purification methods used were optimal and suggest a possible synthetic path of EGT in P. citrinopileatus, which will help better explore the application of EGT.]

Comments 2: Botanical names should be written in italics.

Response 2: Thank you for pointing this out. We agree with this comment. According to your comments, we have revised the botanical name of the whole paper. ‘Pleurotus citrinopileatus Singer’ as been replaced by ‘Pleurotus citrinopileatus Singer’.

Introduction

Comments 3: The introduction provides a good background on the subject of the study but it should be expanded with the addition of a paragraph regarding the application of HPPE in other plants and other mushroom species for the extraction of bioactive ingredients.

Response 3: Thank you for pointing this out. We agree with this comment. According to your comments, we expanded with the addition of a paragraph regarding the application of HPPE in other plants and other mushroom species for the extraction of bioactive ingredients. Lines 59-66. [For example, natural melanin has been extracted from the wild mushroom Auricularia auricula [19], bioactive peptides have been extracted from mushrooms [20], and phenolic compounds have been extracted from various plants [18]. Moreira [21] showed that HHPE is more effective than atmospheric pressure extraction, with an increase in an-tioxidant activity measured by fluorescence recovery after photobleaching FRAP, DPPH, and ABTS assays (29%, 48%, and 70%) and an increase of approximately 40% for all compounds. HHP increases the extraction rate of bioactive compounds and can be performed in a relatively short period.]

Comments 4: The objectives of the study are clear.

Response 4: Thank you for your recognition of the purpose of the study.

Comments 5: Botanical names should be written in italics.

Response 5: Thank you for pointing this out. We agree with this comment. According to your comments, we have revised the botanical name of the whole paper. ‘Pleurotus citrinopileatus Singer’ as been replaced by ‘Pleurotus citrinopileatus Singer’.

Materials and methods

Comments 6: Some additional details would help readers reproduce the work. Proper experimental replication for each analysis should be explicitly stated. Preparation of plant material: Please indicate briefly how mushrooms were broken down and the storage conditions of mushroom powder after drying until further analysis.

Response 6: Thank you for pointing this out. We agree with this comment. According to your comments, we explained it in detail. Lines 93-111. [Natural ventilation drying (ND): …The mushroom powder was sealed and stored in a cool and dry place for further use.

Freeze drying (FD): … The FD mushrooms were then broken up using a grinder (IKA MuItiDrive BASIC S025, IKA (Guangzhou) Instrument Equipment Co., Ltd, China) and sifted through a 50-mesh screen. The mushroom powder was sealed and stored in a cool and dry place for further use.

Hot-air drying (HD): …The HD mushrooms were then broken up using a grinder (IKA MuItiDrive BASIC S025, IKA (Guangzhou) Instrument Equipment Co., Ltd, China) and sifted through a 50-mesh screen. The mushroom powders were sealed and stored in a cool and dry place for further use.]

Comments 7: Lines 82, 92: Authors should indicate how much time has taken, approximately, for drying in the oven and with natural ventilation.

Response 7: Thank you for pointing this out. We agree with this comment. According to your comments, we explained it in detail. Lines 94, 101, 108. [dry naturally for 72 h until it reached a constant weight. for freeze-drying for 24 h, and dried for 24 h until a constant weight was achieved.]

Comments 8: Line 87. Please provide the model and manufacturer of the vacuum pressure.

Response 8: Thank you for pointing this out. We agree with this comment. According to your comments, we explained it in detail. Lines 100-101. [freeze-drying machine (Pilot7-12E, BIOCOOL Instrument Co., Ltd, Beijing, China) for freeze-drying for 24 h,]

Comments 9: Line 91. Please provide the model and manufacturer of the oven

Response 9: Thank you for pointing this out. We agree with this comment. According to your comments, we explained it in detail. Line 107. [oven (DHG-9015A, Yiheng Scientific Instrument Co., Ltd, Shanghai, China)]

Comments 10: Paragraph 2.2.2 amino acids analysis

Please provide a detailed description for the protocol, the instruments used and the whole sample preparation (eg. type of membrane filter), since you reference someone else’s method.

Response 10: Thank you for pointing this out. We agree with this comment. According to your comments, we explained it in detail. Lines 113-122. [The change in amino acid content in the mushrooms in different states (fresh, ND, FD, and HD) was determined, and its influence on EGT content was analyzed [22]. A specific amount of mushroom in the different state (fresh, ND, FD, and HD) was weighed into the hydrolysis tubes, mixed with 10 mL 1:1 hydrochloric acid solution, and placed in a 110 °C oven for hydrolysis for 22 h. Following hydrolysis, the samples were cooled, transferred to a 10 mL volume bottle and made up to a final volume of 10 mL with 0.02 mol/L HCL. The sample solution was filtered through a 0.22 μm membrane. Then, an aliquot of 20 μL of the mixed amino acid standard working solution and sample determination solution was injected into the amino acid analyzer (LA8080, HITACHI Co., Ltd, Japan).]

Comments11: Paragraph 2.2.5 EGT content determination

Please provide a detailed description for the protocol, the HPLC system used and the whole sample preparation, since you reference someone else’s method.

Response11: Thank you for pointing this out. We agree with this comment. According to your comments, we explained it in detail. Lines 141-153. [The contents of EGT in mushrooms in different states (Fresh, ND, FD, and HD) were determined. The moisture content of mushrooms in different states (Fresh, ND, FD, and HD) was determined by a rapid moisture meter (HX204, METTLER TOLEDO Technology Co. Ltd, China). Subsequently, the samples were treated according to the existing method [25]. The fresh mushrooms or mushroom powder (ND, FD, and HD) mixed with 60% ethanol at a solid-to-liquid ratio of 1:30 (w/v). The solid-liquid mixture was extracted in a water bath maintained at a constant temperature of 70 °C for 1 h and centrifuged at 9500 ×g for 10 min. The supernatant was collected, and the volume was made up to 100 mL with 60% ethanol to determine the amount of EGT extracted by HPLC (1260 II Prime, Agilent Technologies Inc., USA) [25].

HPLC detection conditions: The column (XBridge® BHE Amide 5 μm) was manu-factured by Agilent Technologies Inc. The mobile phase was 98:2 water and methanol, flow rate was 1 mL/min, sample volume was 20 μL, and detection wavelength was 254 nm.]

Comments 12: Please provide the manufacturers of the assay kit used for the measurement of antioxidant activity.

Response 12: Thank you for pointing this out. We agree with this comment. According to your comments, we explained it in detail. Lines 213, 221, 229. [(Yuanye Bio-Technology Co., Ltd., Shanghai, China)]

Results and discussion

Comments 13: Figures and Tables: abbreviations should be explained in the legend

Response 13: Thank you for pointing this out. We agree with this comment. According to your comments, we explained the abbreviations in the legend. Line 295. []

Comments 14: Please provide the water content of the raw mushrooms.

Response 14: Thank you for pointing this out. We agree with this comment. According to your comments, we explained it in detail. Lines 142-144. [The moisture content of mushrooms in different states (Fresh, ND, FD, and HD) was determined by a rapid moisture meter (HX204, METTLER TOLEDO Technology Co. Ltd, China), and were 90.32%, 0.77%, 0.52, and 0.56%, respectively.]

Comments 15: Figure 2. The content of EGT is expressed as mg/g? Shouldn’t rather be expressed as mg/g d.w.? Same at Table 2.

Response 15: Thank you for pointing this out. We agree with this comment. According to your comments, we have revised the whole text. ‘mg/g’ as been replaced by ‘mg/g d.w.’

Comments 16: Table 2. Please provide the measuring unit in the legend

Response 16: Thank you for pointing this out. The measuring unit in the legend is %. Table 2. [Amino acids (%)].

Comments 17: Table 4. Please correct residual instead of Residua and provide an explanation for R2 R2Adj, and R2 pred.

Response 17: Thank you for pointing this out. We agree with this comment. According to your comments, we have revised ‘residual’. And provided an explanation for R2 R2Adj, and R2 pred. Lines 371-376. [The effectiveness of EGT extraction is better when the F-value is larger. R2 indicates the degree of model fit and the closer R2 is to 1, the better the fit. R2Adj indicates that R2 is modified to avoid overfitting. R2pred means the degree to which the model predicts the response of new observations. Models with larger predictive R2 values also have better predictive power. The gap between R2Adj and R2pred should be less than 0.2]

Comments 18: Statistical results should be provided between predicted values and experimental values.

Response 18: Thank you for pointing this out. We agree with this comment. According to your comments, we explained it in detail. Lines 386-387. [the measured EGT yield obtained by HHPE (4.03 ± 0.01 mg/g d.w. P > 0.05) was no significant difference from the predicted value,]

Comments 19: Line 329: Please correct “anthocyanins”

Response 19: Thank you for pointing this out. We agree with this comment. We corrected the ‘anthocyanin’ to EGT. Line 390.

Conclusions

Conclusions highlight the most novel outcomes and implications of this study.

References

Comments 20: References are not appropriately cited, so this section should be thoroughly and carefully double-checked and revised to follow the standard references style of the journal.

Response 20: Thank you for pointing this out. We agree with this comment. We checked the references and made sure they were all quoted correctly. Lines 211.

Round 2

Reviewer 2 Report

Comments and Suggestions for Authors

All comments were taken into account and the authors made appropriate corrections. Therefore, it is accepted for publication

Reviewer 3 Report

Comments and Suggestions for Authors

The manuscript has been improved considerably, and I propose that it be accepted for publication. However, the English language should be checked (e.g., line 312) throughout the manuscript.

Comments on the Quality of English Language

Moderate editing of English language is required.